# Distinct immune microenvironment profiles of therapeutic responders emerge in combined TGFβ/PD-L1 blockade-treated squamous cell carcinoma

Alexander A. Strait [1], Rachel A. Woolaver[2], Spencer C. Hall [1], Christian D. Young[1], Sana D. Karam[3], Antonio Jimeno[4], Yan Lan[5,6], David Raben[3], Jing H. Wang [2,8✉] & Xiao-Jing Wang [1,7✉]

Transforming growth factor beta (TGFβ) and programmed death-ligand 1 (PD-L1) are often overproduced in refractory squamous cell carcinoma (SCC). We examined spatial patterns of PD-L1[+] cells in mouse and human SCCs and found that PD-L1 was primarily expressed on infiltrating leukocytes. Although combined TGFβ and PD-L1 blockade are undergoing cancer clinical trials, there are no predictive markers for therapeutic responders. To address this, we used both a small molecule TGFβ inhibitor in combination with anti-PD-L1 and a bifunctional fusion protein targeting both TGFβ and PD-L1 to treat mouse SCCs and found TGFβ inhibition enhanced PD-L1 blockade-induced tumor eradication in multiple tumor models. Furthermore, we identified distinct cell populations of responders and non-responders to bintrafusp alfa, with responders showing a shift toward a more immune-permissive microenvironment. The cellular and molecular signatures of responders *versus* non-responders to combined TGFβ and PD-L1 blockade provide important insights into future personalized immunotherapy in SCC.

[1] Department of Pathology, University of Colorado, Anschutz Medical Campus, Aurora, CO, USA. [2] Department of Immunology and Microbiology, University of Colorado, Anschutz Medical Campus, Aurora, CO, USA. [3] Department of Radiation Oncology, University of Colorado, Anschutz Medical Campus, Aurora, CO, USA. [4] Department of Medicine, University of Colorado, Anschutz Medical Campus, Aurora, CO, USA. [5] EMD Serono Research and Development Institute Inc., Billerica, MA, USA. [6] a business of Merck KGaA, Darmstadt, Germany. [7] Veterans Affairs Medical Center, VA Eastern Colorado Health Care System, Aurora, CO, USA. [8] Present address: Hillman Cancer Center, Department of Medicine, University of Pittsburgh, Pittsburgh, PA, USA. ✉email: jhw51@pitt.edu; xj.wang@cuanschutz.edu

Squamous cell carcinomas (SCCs) arise from stratified epithelia and are associated with environmental carcinogens; skin SCC is associated with UV irradiation and head and neck SCC (HNSCC) is associated with tobacco carcinogens or human papilloma virus (HPV) infection[1]. Although its incidence has decreased, its mortality rate has increased since 2012[1], indicating that treatment options for patients have stagnated and novel therapeutics have not yet had a dramatic impact on patient outcomes. This is even more pronounced in HPV-negative HNSCC, which is associated with worse prognoses and is less immunogenic[2,3]. SCC has a high mutational burden[4] and HPV-negative patients commonly score positive for programmed death-ligand 1 (PD-L1) expression[5], both of which are predictors of immunotherapy response. However, despite FDA approval for programmed cell death protein 1 (PD-1) and PD-L1 blockade in recurrent and metastatic HNSCC patients, the response rate remains between 10 and 20%[6,7], suggesting that additional immunosuppressive mechanisms need to be targeted and that immunotherapy treatments need to be tailored for individual patients in order to improve their effectiveness.

SCC is commonly associated with dysregulated TGFβ signaling, with 69% of human patients exhibiting *TGFβRII* downregulation and 35% exhibiting SMAD4 deletions in tumor epithelial cells, which is critical for canonical TGFβ signaling[8,9]. In contrast, the TGFβ ligand is often overexpressed in skin SCCs and HNSCCs with the loss of these TGFβ signaling components[10,11]. As tumors progress and lose the capacity to respond to TGFβ-mediated growth arrest effect on tumor epithelial cells, TGFβ promotes tumor progression via numerous pathways[12], including functioning as a potent immune suppressor[13], thus presenting itself as a potential therapeutic target in locally advanced SCCs. Recently, TGFβ has emerged as a promising treatment target in combination with PD-1/PD-L1 blockade in multiple tumor models; TGFβ signaling in fibroblasts prevents cytotoxic T lymphocyte (CTL) infiltration into tumors, while PD-L1 expression in the tumors inhibits effector T cell function[14–17]. Other studies have shown a link between TGFβ and PD-L1 signaling in cancer; TGFβ regulates PD-L1 expression in vitro in a Smad2-dependent manner[18], and it also regulates PD-1 expression on tumor-infiltrating T cells[19]. PD-1 blockade also activates tumorigenic TGFβ signaling in tumor cells that can be targeted by a TGFβ depleting antibody to enhance the immune checkpoint blockade response[20], and conversely TGFβ-induced EMT can promote PD-L1 expression[21]. This has led to the development of bintrafusp alfa, a bifunctional fusion protein composed of the extracellular domain of the TGF-βRII receptor (a TGF-β "trap") fused to a human IgG1 mAb blocking PD-L1, which has shown encouraging preclinical and clinical antitumor activity[22–25]. To date, preclinical studies of bintrafusp alfa have focused primarily on the activation of CD8+ T cells and NK cells in the tumor microenvironment (TME) of tumors derived from murine breast and colorectal cell lines, but its effectiveness on the poorly immunogenic microenvironment associated with SCC has not been shown. In other cancer types, much of the research on the effectiveness of immune checkpoint blockade has focused on its effects on T cell function in the immune microenvironment[26–28]. However, recent studies have suggested infiltrating myeloid cells are critical for the response to PD-L1 blockade[29,30], although this has not been studied in the context of SCC.

To address challenges in immunotherapeutic strategies for SCC, we previously developed a genetic mouse model of SCC that recapitulates the initiation and progression of the most aggressive forms of HPV− SCC; genetic deletion of Smad4 in combination with an activating Kras mutation in K15+ stem cells results in spontaneous oral and skin tumors with elevated TGFβ secretion[11]. From these we derived a SCC cell line syngeneic to

C57BL/6 recipients[11,31,32] and used it as a tumor transplant model together with additional syngeneic transplant SCC models to assess if combined TGFβ inhibition and PD-L1 blockade induces durable tumor eradication. We used high-dimensional analysis of cell surface markers and single-cell RNA sequencing to identify immune cell types and factors in the TME that affect the ability of CTLs to clear tumor cells. We also identify distinct cellular and molecular signatures of responders and non-responders that are unique to treatment with combined TGFβ inhibition and PD-L1 blockade, and demonstrate their requirement for a successful anti-tumor response to bintrafusp alfa.

## Results

**Secreted TGFβ functions as an immune suppressor in the SCC TME, and combined TGFβ inhibition and PD-L1 blockade durably eradicated these tumors.** We have previously shown elevated TGFβ levels in SCCs with a Smad4 deletion[11]. A223, a cell line derived from these tumors, also exhibited elevated TGFβ levels in vitro compared to untransformed mouse skin keratinocytes (MNK) and the murine HNSCC cell lines LY2[33], and A1419, a SCC cell line we derived from 4NQO-induced SCC ($P < 0.0001$ for all, $n = 6$) (Fig. 1a). We confirmed the deletion of Smad4 in A223 cells relative to A1419, LY2, and control HaCaT keratinocytes by western blot (Supplementary Fig. 1a). Furthermore, an analysis of 530 human HNSCC tumors (TCGA, Firehose Head and Neck) using The Cancer Genome Atlas via cBioportal[34,35] shows increased TGFB1 mRNA levels in SMAD4-deleted tumors ($n = 253$) relative to SMAD4 diploid tumors ($P = 0.0011$, $n = 205$), and no change in tumors with SMAD4 gain ($P = 0.4215$, $n = 30$) (Fig. 1b). Because TGFβ functions as a tumor promoter as tumor cells become unresponsive to TGFβ-induced growth arrest, we hypothesized that paracrine effects of TGFβ in A223 tumors could be targeted by TGFβ inhibition. The small-molecule TGFβ inhibitor LY2109761 effectively inhibits canonical TGFβ signaling in A223 tumors as measured by phospho-Smad3 immunohistochemistry (IHC) and quantified in QuPath software ($P = 0.0287$, $n = 5$) (Supplementary Fig. 1b, c). When A223 recipient mice were treated with LY2109761 from the time of s.c. tumor cell transplantation into syngeneic C57BL/6 mice, tumor growth was modestly inhibited in recipients ($P = 0.0196$) (Fig. 1c, d), but not in athymic nude mice (Fig. 1e, f). This indicates that the antitumor effect of TGFβ inhibition was on the immune microenvironment rather than intrinsic to the tumor cells; thus, the adaptive immune response plays a key role in controlling the growth of A223 tumors. However, treatment from the time of tumor initiation does not mimic clinical treatment of advanced SCC. Instead, we sought to assess the effect of TGFβ inhibition as an adjunct to immunotherapy. We combined LY2109761 with anti-PD-L1 and found that TGFβ inhibition increased the rate of tumor eradication from 33% with anti-PD-L1 alone to 77% with combination therapy and further reduced tumor growth relative to single agents or control alone (Fig. 1g, h). In addition, when surviving mice ($n = 7$ bintrafusp alfa-treated and $n = 3$ anti-PD-L1-treated) were rechallenged with A223 cells s.c. 147 days after initial tumor cell transplantation, none grew tumors within 80 days, while a cohort of naïve mouse recipients rapidly developed tumors and reached survival endpoints within 30 days (Fig. 1g). These data suggest that systemic TGFβ inhibition improved the effectiveness of T cell-mediated immunotherapy and combined treatment induced a memory T cell response against A223 tumors.

**PD-L1 expression in A223 tumors is primarily on CD11b+ myeloid cells and not induced by TGFβ.** To identify the mechanism of how TGFβ signaling impedes the response to anti-

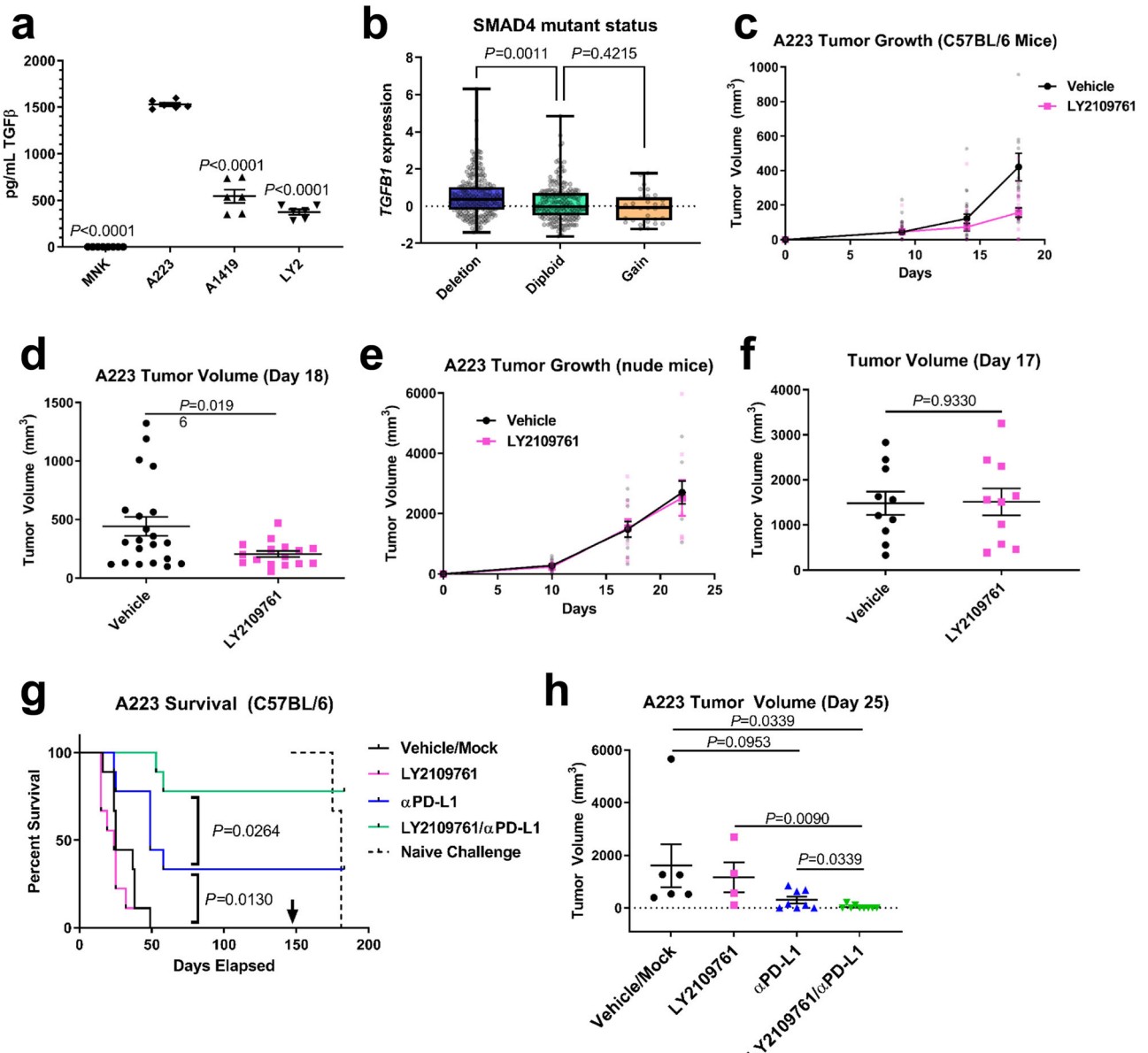

**Fig. 1 Smad4-mutant SCCs in mice and humans exhibit high TGFβ expression, and combined TGFβ inhibition and PD-L1 blockade durably eradicate Smad4-mutant (A223) SCCs. a** pg/mL per cell of secreted TGFβ measured by ELISA in untransformed mouse skin keratinocytes, A223 cells, LY2 cells, and A1419 cells ($n = 6$). P values are shown for multiple comparisons to A223 cells by one-way ANOVA ($F = 328.8$, $df = 25$). **b** *TGFB1* mRNA expression z-scores in human HNSCC patient samples based on their *SMAD4* mutant status of deleted (blue, $n = 253$), diploid (green, $n = 205$), or gained (orange, $n = 30$). P values are shown for multiple comparisons to diploid cells by one-way ANOVA ($F = 8.445$, $df = 487$). **c** Post-transplant tumor volume of A223 tumors in C57BL/6 mice treated with either 75 mg/kg LY2109761 p.o. per day (pink, $n = 22$), or carboxymethylcellulose vehicle control (black, $n = 22$) for 14 days. **d** Tumor volume on day 18 post-transplant of mice in (**c**). **e** Post-transplant tumor volume of A223 in athymic nude mice treated as in (**c**) ($n = 10$ per treatment group). **f** Tumor volume on day 17 post-transplant of mice in (**e**). **g** Survival post-injection of C57BL/6 mice bearing A223 tumors that were treated as in (**c**), with the addition of 200 μg anti-PD-L1 or its isotype control 3×/week for 2 weeks (vehicle/mock: black lines, LY2109761/mock: pink lines, vehicle/anti-PD-L1: blue lines, LY2109761/anti-PD-L1: green lines, $n = 9$ per treatment group); surviving mice (combination-treated $n = 7$, anti-PD-L1-treated $n = 3$) were rechallenged with A223 cells on day 147 (arrow) along with a cohort of naïve mice (dashed line, $n = 5$). **h** Tumor volume on day 25 post-transplant in mice from (**g**). Multiple comparisons were corrected with Tukey's method for (**a**) and (**b**). Unpaired two-tailed t tests were performed for (**d**), (**f**), and (**h**), and survival comparisons were performed using the log-rank Mantel–Cox test. All error bars represent the SEM, except for **b** where the box represents the 25–75th percentile, whiskers represent the full range, and the line represents the median value.

PD-L1, we explored the relationship between TGFβ signaling and PD-L1 expression in SCC. In concordance with previous research, recombinant TGFβ induced PD-L1 expression in LY2 cells, a murine model of carcinogen-induced HNSCC with an intact TGFβ signaling pathway, in a Smad4-dependent manner. The addition of recombinant TGFβ to the growth media in vitro significantly increased PD-L1 surface protein expression

($P = 0.0103$, $n = 4$, gating strategy in Supplementary Fig. 2) (Fig. 2a), and reduced protein expression of Smad4 by siRNA diminished PD-L1 expression in LY2 cells (Supplementary Fig. 3a, b). This suggests that the TGFβ-induced expression of PD-L1 on LY2 tumor cells is dependent on Smad4-mediated canonical TGFβ signaling. However, A223 cells lacked PD-L1 expression in vitro both innately and in response to TGFβ

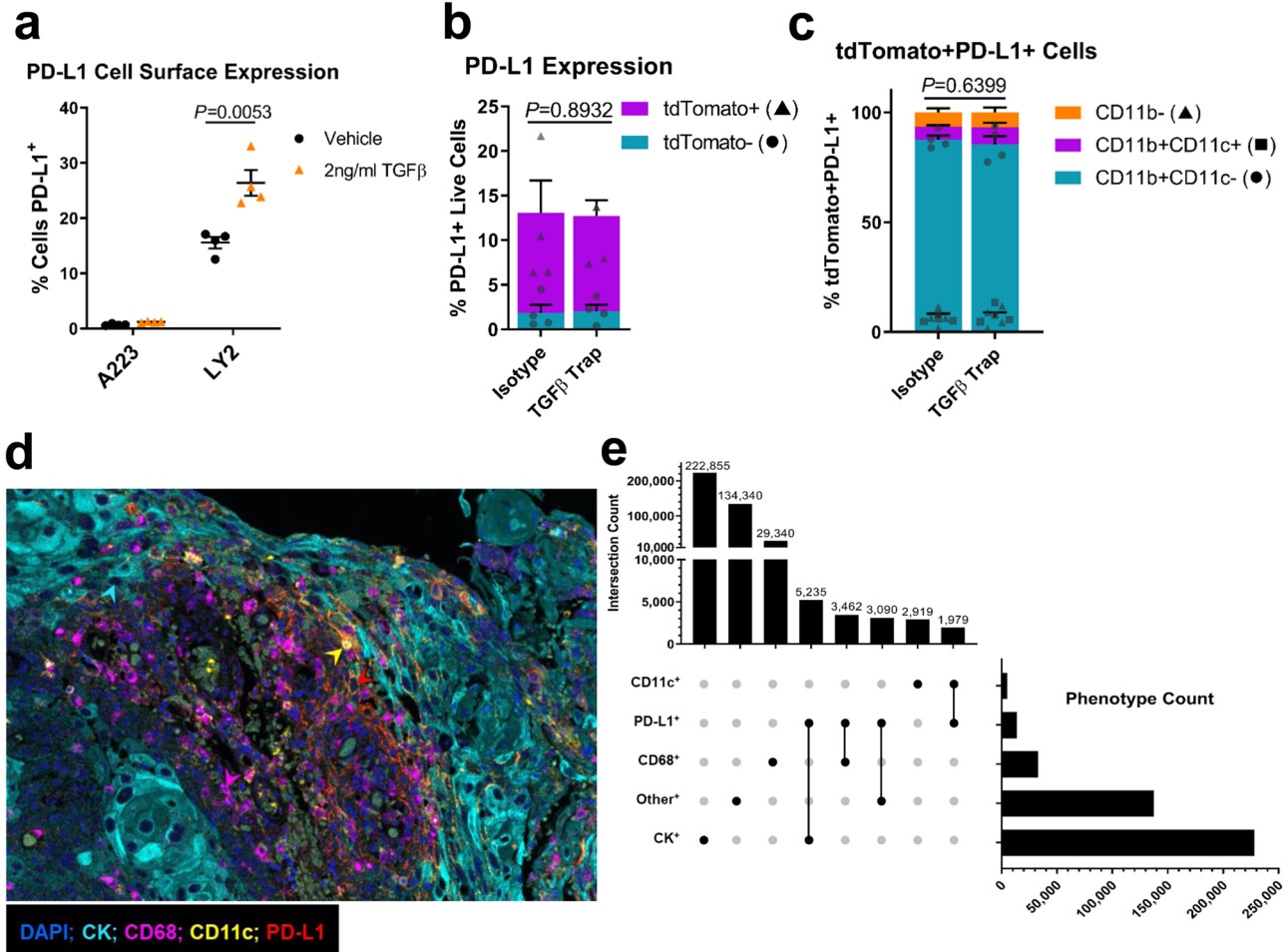

**Fig. 2 Spatial patterns of PD-L1+ cells in mouse and human SCCs demonstrate that infiltrating leukocytes are the primary source of PD-L1 within tumors. a** Percentage of PD-L1+ A223 and LY2 cells after stimulation with 2 ng/ml recombinant TGFβ (orange triangles, $n = 4$) or PBS vehicle (black circles, $n = 4$) as measured by flow cytometry. **b** tdTomato expression (purple bars: positive, blue bars: negative) in PD-L1+ live cells isolated from mTmG mice bearing A223 tumors and treated with TGFβ trap control isotype control 3×/week for 1 week, as quantified by flow cytometry ($n = 5$ per treatment group). **c** Analysis of CD11b and CD11c expression of tdTomato+ cells from **c** as quantified by flow cytometry (CD11b+ CD11c−: blue bars, CD11b+ CD11c+: purple bars, CD11b−: orange bars, $n = 5$ per treatment group). **d** Representative tumor from multispectral imaging analysis of 23 human HNSCC tumors stained for pan-cytokeratin (cyan), CD68 (magenta), CD11c (yellow), PD-L1 (red), and DAPI (blue). Arrowheads represent example phenotypes using the same color code. **e** Table of total number of cells of each indicated phenotype across all samples in (**d**) (horizontal bars), and counts of cells that overlap one or more phenotypes (vertical bars). The central matrix displays the combinations graphically with connected dots indicating overlapping phenotypes. Unpaired two-tailed *t* tests were performed for (**a**–**c**), and all error bars represent the SEM. Raw unstacked values in (**b**) and (**c**) are represented by the indicated symbols.

stimulation (Fig. 2a), suggesting that their in vivo tumors could be PD-L1-negative. To test this, we transplanted A223 cells s.c. into mTmG recipient mice that constitutively express tdTomato in the absence of Cre recombinase in order to differentiate between PD-L1 protein expression on the tdTomato− A223 tumor cells and tdTomato+ recipient cells in the TME. Interestingly, A223 tumors were PD-L1+ (13.09% of tumor cells); however, the PD-L1+ cells were primarily tdTomato+ recipient cells (85.88%), and the majority of those cells (87.46%) were CD11b+ CD11c− myeloid cells (Fig. 2b, c, gating strategy in Supplementary Fig. 4). Furthermore, a TGFβ trap that neutralizes secreted TGFβ (described below) had no effect on the number of PD-L1+ cells within the tumor, indicating that the A223 PD-L1+ TME is not maintained by TGFβ signaling. We then analyzed 23 human HNSCC samples to determine whether they had a similar distribution of PD-L1+ cells in PD-L1-positive tumors. We used multispectral imaging to assess cytokeratin (CK), CD68, CD11c, and PD-L1 expression on each cell and phenotyped cells both as PD-L1+ or PD-L1− cells, and as CK+ tumor cells, CD68+ macrophages, CD11c+ dendritic cells, or CK−CD68−CD11c−

other stromal cells, (Fig. 2d, Supplementary Fig. 3c–f). Although a plurality of PD-L1+ cells overlapped with CK+ cells, we found that the majority of PD-L1+ phenotypes overlapped with CD68+, CD11c+, and other stromal cells (Fig. 2e). In addition, we calculated the relative intensity of PD-L1 expression on each cell type and found that the majority of PD-L1+ cells in each individual tumor expressing PD-L1 were CK− cells in the TME, with CD11c+ dendritic cells having the highest PD-L1 expression per cell despite being less prevalent than CD68+ macrophages (Supplementary Fig. 3g). Taken together, these results indicate that A223 tumors do not express PD-L1 in a TGFβ-dependent manner, and the primary target for anti-PD-L1 in both A223 and human HNSCC tumors lies on infiltrating leukocytes rather than the tumor cells themselves.

**Bintrafusp alfa, a bifunctional fusion protein targeting TGFβ and PD-L1, durably eradicates mouse SCCs and produces distinct responder and nonresponder phenotypes.** We used bintrafusp alfa (M7824), a bifunctional fusion protein targeting TGFβ and PD-L1, to co-localize TGFβ sequestration and PD-L1

blockade to the A223 TME in contrast to systemic inhibition of TGFβ receptor activity by the small molecule LY2109761. We treated mice bearing A223 tumors with bintrafusp alfa or its single target controls: anti-PD-L1 (Avelumab), TGFβ trap control (bintrafusp alfa with a mutated PD-L1 binding domain), and an isotype control (anti-PD-L1 with a mutated PD-L1 binding site). IHC staining against pSmad3 confirmed that the TGFβ trap control ($P = 0.0034$) and bintrafusp alfa ($P = 0.0462$) inhibited canonical TGFβ signaling relative to the isotype control (Supplementary Fig. 5a). Similarly to the combination of LY2109761 and anti-PD-L1, bintrafusp alfa ($n = 12$) significantly improved survival relative to the isotype control ($n = 8$) and TGFβ blockade ($n = 8$) alone ($P = 0.0035$), and to PD-L1 blockade alone ($P = 0.0438$, $n = 10$) (Fig. 3a). Tumor size was significantly lower in the bintrafusp alfa-treated mice by day 23 ($P = 0.0129$, $n = 10–12$) (Fig. 3b), and mice that responded to treatment rapidly cleared their tumors and remained tumor-free for the duration of the experiment (Fig. 3c–f). We then rechallenged surviving mice (5/12 bintrafusp alfa-treated and 1/10 anti-PD-L1-treated) with A223 tumor cells s.c. 124 days after initial tumor cell transplant, and none of them had detectable tumor growth within 100 days while all naïve mouse recipients rapidly developed

tumors within 30 days (Fig. 3g), indicating that surviving mice maintain a durable memory T cell response against A223 cells. Notably, when A223 tumors in CD8-knockout mouse recipients received the above treatments, none of the treatments had any effect on survival (Fig. 3h) or tumor growth (Fig. 3i), implying that the effect of bintrafusp alfa was ultimately dependent on effector T cell-mediated tumor clearance. In addition, bintrafusp alfa reduced tumor growth in Balb/c mice bearing syngeneic LY2 tumors (Supplementary Fig. 5b), and in C57BL/6 mice bearing syngeneic tumors from the 4-NQO-driven SCC cell line A1419 (Supplementary Fig. 5c), indicating that the effectiveness of this combination in SCC is not limited to a single SCC genotype. Taken together, this suggests that mice respond to bintrafusp alfa in a CD8 T cell-dependent manner that is conserved across multiple SCC models, and that they maintain a sustained memory T cell response against the tumor cells.

**Responders to bintrafusp alfa have a more immune-permissive microenvironment and increased T-cell activation when compared to nonresponders.** We found that tumor-bearing mice had no partial responses to bintrafusp alfa; surviving mice either had complete tumor clearance or rapidly reached endpoints. Thus, we

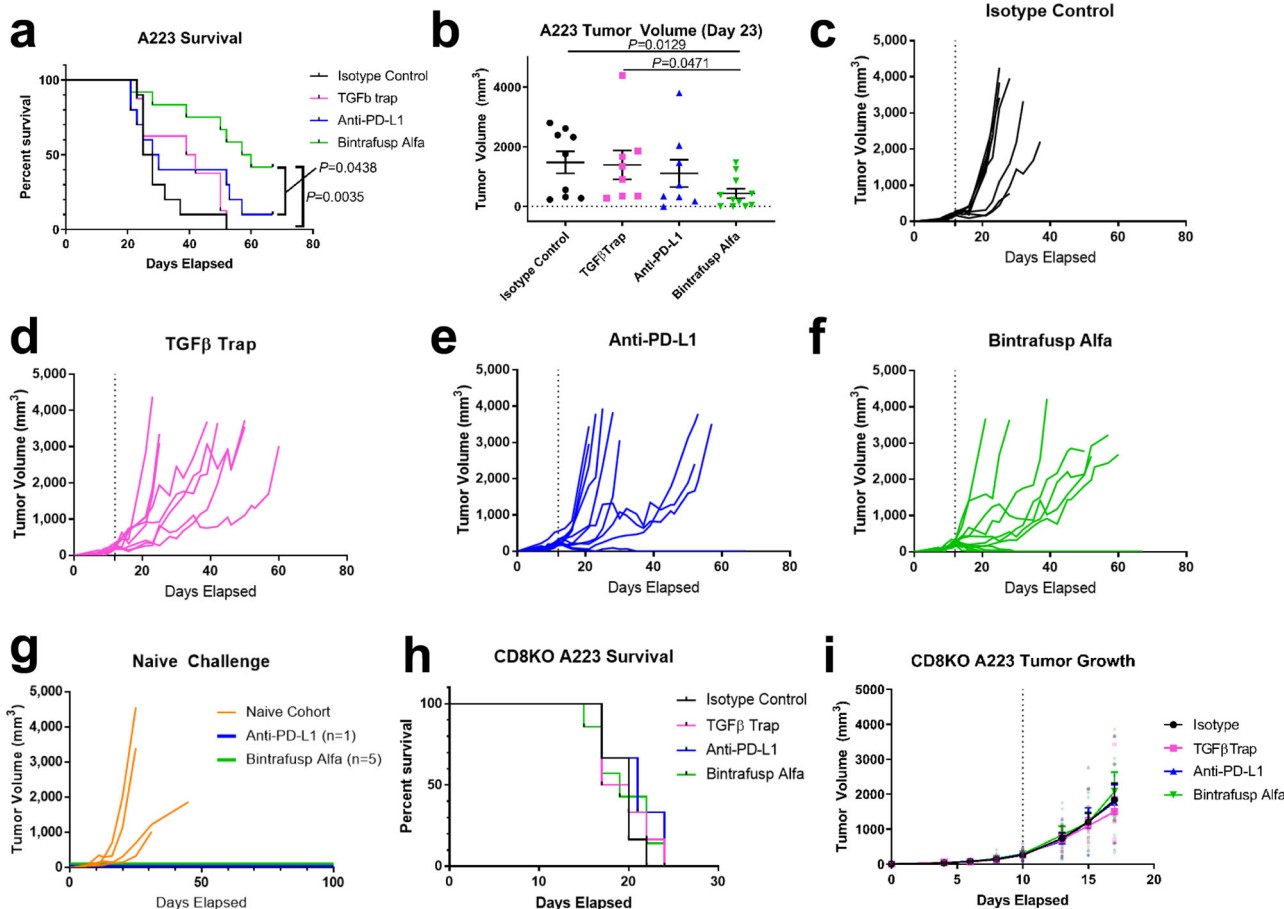

**Fig. 3 Bintrafusp alfa, a bifunctional fusion protein targeting TGFβ and PD-L1, durably eradicates A223 tumors and produces distinct responder and nonresponder phenotypes. a** Survival post-injection of C57BL/6 mice bearing A223 tumors that were treated with isotype control ($n = 8$), TGFβ trap control ($n = 8$), anti-PD-L1 ($n = 10$), or bintrafusp alfa ($n = 12$) 3×/week for 2 weeks. **b** Tumor volume on day 23 post-transplant for mice in (**b**). **c** Individual tumor growth for isotype-treated mice in (**a**). **d** Individual tumor growth for TGFβ trap-treated mice in (**a**). **e** Individual tumor growth for anti-PD-L1-treated mice in (**a**). **f** Individual tumor growth for bintrafusp alfa-treated mice in (**a**). **g** Surviving mice from (**a**) ($n = 1$ anti-PD-L1-treated and $n = 5$ bintrafusp alfa-treated) were rechallenged 124 days post-transplant tumor growth as indicated; orange lines represent tumor growth of naïve mice injected with A223 cells simultaneously ($n = 5$). Dashed lines for **c–g** indicate the start of treatment. **h** Survival of syngeneic CD8-knockout mice (CD8KO) injected with A223 cells and treated as in (**a**) ($n = 7$ bintrafusp alfa-treated, $n = 6$ for all other treatments). **i** Tumor growth of mice from (**h**). Unpaired two-tailed $t$ tests were performed for (**b**) and survival comparisons were performed in (**a**) and (**d**) using the log-rank Mantel–Cox test, and all error bars represent the SEM.

are able to differentiate mice treated with bintrafusp alfa into distinct groups of responders with regressing tumors and non-responders with growing tumors for cellular and molecular analyses. We used CyTOF mass cytometry to profile cells based on their expression of 22 protein markers (Supplementary Tables 1 and 2, Supplementary Fig. 6a, b, gating strategy in Supplementary Fig. 7). We found that CD8 T cell infiltration relative to the isotype control was increased by bintrafusp alfa ($P < 0.0001$), the TGFβ trap control ($P = 0.0018$), and anti-PD-L1 ($P = 0.0175$) (Fig. 4a). In addition, the TGFβ trap control decreased M2 macrophage infiltration relative to the isotype control ($P < 0.0001$) and increased resident monocyte infiltration relative to bintrafusp alfa ($P = 0.0129$) (Fig. 4a). However, the lack of an effect on tumor growth by the TGFβ inhibitor alone suggested that its effect on T cell infiltration was insufficient for tumor eradication without the addition of a PD-L1 blockade. Additionally. when we stimulated tumor-infiltrating T cells ex vivo with ionomycin and PMA, there were no changes in TNFα⁺IFNγ⁺ cells relative to the isotype control (Supplementary Fig. 8a), although CD8 T cells from bintrafusp alfa-treated tumors had elevated granzyme B levels relative to isotype ($P = 0.0064$) and anti-PD-L1- ($P = 0.0120$) treated tumors (Supplementary Fig. 8b). These data indicate that although bintrafusp alfa promoted T cell activation, the variability in its response makes it difficult to pinpoint the mechanism of its effect on infiltrating immune cells.

To better identify the therapeutic effects of bintrafusp alfa we chose to focus specifically on responders to therapy in order to characterize changes in immune cells responsible for an antitumor immune response. We identified four days after the start of treatment as the earliest point when we could separate bintrafusp alfa-treated mice into statistically distinct group of responders (<600 mm³) and nonresponders (>600 mm³) (Supplementary Fig. 8c, g), and we found significant differences between tumor-infiltrating immune cell populations ($P < 0.0001$) (Fig. 4b). Responder tumors had significantly decreased infiltration of resident monocytes ($P = 0.0199$), which are associated with an immune-suppressive TME (Fig. 4b)[36,37]. They also exhibited significantly increased infiltration of CD8 T cells ($P < 0.0001$) and M1 macrophages ($P = 0.0062$) (Fig. 4b) which are indicative of an immune-permissive TME. Infiltrating CTLs in responder tumors had elevated levels of PD-1 expression ($P = 0.0332$), further suggesting increased T cell activation (Fig. 4c). Bintrafusp alfa responders also had a greater proportion of infiltrating CD8⁺ T cells cultured ex vivo that were TNFα⁺ IFNγ⁺ ($P = 0.0003$) and granzyme B⁺ ($P = 0.0021$) in response to ionomycin and PMA stimulation (Fig. 4d, e), indicating that they had increased potential to activate cytotoxic activity.

To assess whether these population changes were specific to tumor size rather than bintrafusp alfa treatment, we also separated isotype control-treated tumors into small (<600 mm³) and large (>600 mm³) size groups where they showed significant divergence in tumor size (Supplementary Fig. 8d, g) and found no significant differences between infiltrating cell populations (Supplementary Fig. 9a). Tumors treated with TGFβ trap or anti-PD-L1 alone had a similar lack of change between size groups (Supplementary Fig. 8e–g, Supplementary Fig. 9a, b). In addition to tumor size, we also measured mouse weight during treatment until the point of the analyses in Fig. 4 and did not find weight differences among groups (Supplementary Fig. 8h), suggesting that metabolic changes, if any, are insufficient to contribute to bintrafusp alfa-mediated tumor eradication. Taken together, these results show that the therapeutic effect of bintrafusp alfa is specific to combined TGFβ and PD-L1 inhibition rather than an effect of tumor size alone. In addition,

the presence of responder and nonresponder subpopulations mask the overall treatment effect of bintrafusp alfa on infiltrating immune cell populations; bintrafusp alfa responders have a unique profile of infiltrating immune cells indicative of an antitumor immune microenvironment.

**Single-cell RNA sequencing shows that responders have unique conserved signaling pathways that promote antigen presentation and myeloid and T cell infiltration.** To confirm our mass cytometry profiling, we used single-cell RNA sequencing to identify transcriptional changes between bintrafusp alfa responders ($n = 2$) and nonresponders ($n = 2$). After UMAP clustering of cells based on their gene expression, we compared gene expression from each cluster to a list of known markers for various cell types supplemented by additional curated markers where indicated (Supplementary Figs. 10–14); markers for A223 tumor cells were identified from RNA sequencing of A223 cells in vitro and are listed in Supplementary Fig. 15a. UMAP Clusters representing proliferating cells were identified both by their characteristic ring shape (Fig. 5a) and by gene set representing mitotic activity (Supplementary Fig. 15b). We then used the resulting phenotypes of each UMAP cluster to identify distinct populations of infiltrating tumor and stromal cells (Fig. 5a) and split them into responder and non-responder categories based on their tumor growth as defined above, indicating a general shift from tumor cell populations to infiltrating immune cell populations in responders (Supplementary Fig. 16a, b). We quantified that shift by comparing the proportion of each cell population based on their responder status and found that responders had increased levels of infiltrating CD4 T cells, CD8 T cells, and NK cells, while the proportion of tumor cells was considerably lower (Fig. 5b). In addition, responders had elevated proportions of dividing NK and CD8 T cells and a decreased proportion of dividing tumor cells. These results indicate that responders have elevated T and NK cell-mediated antitumor cytotoxic activity, resulting in fewer proliferating tumor cells and reduced tumor growth.

Because we identified changes in infiltrating myeloid cell populations, we sought to identify conserved signaling pathways in responder myeloid cells. When we compared average gene expression of responders vs. nonresponders in macrophages, monocytes, dendritic cells, and tumor cells we found that some of the most over-represented genes in responders were classical and non-classical MHC class I subunit genes H2-Q4, H2-Q6, H2-Q7, H2-K1, H2-D1, H2-T22, and H2-T23, and MHC class II subunit genes CD74, H2-Ab1, H2-Aa, and H2-Eb1, with tumor cells generally lacking class II expression (Fig. 6a). This suggests that the responder TME has increased antigen presentation, which in turn is more permissive of antitumor T cell activity. Because MHC-I downregulation in PD-L1-positive HNSCC tumors is associated with poor prognoses in HNSCC[38], it may also serve as a predictive indicator of the response to bintrafusp alfa.

We also sought to use our single-cell RNA sequencing data to identify chemokine signaling pathways that could be responsible for the remodulation of the immune microenvironment. When we identified differentially expressed genes in responder and nonresponder tumor cells, we found that IL-33 expression was noticeably lower in responder tumor cells (Fig. 6b). IL-33 expression is notable because it promotes tumor growth and metastasis of tumor cells, and it can promote antitumor immunity via its effect on helper T cells[39]. Because its expression was primarily on tumor cells in nonresponders, we infer that it functions as an autocrine tumor promoter in the TME. A major signaling hallmark of responders across cell categories was increased *Stat1* expression (Fig. 6c), indicating a tumor-wide

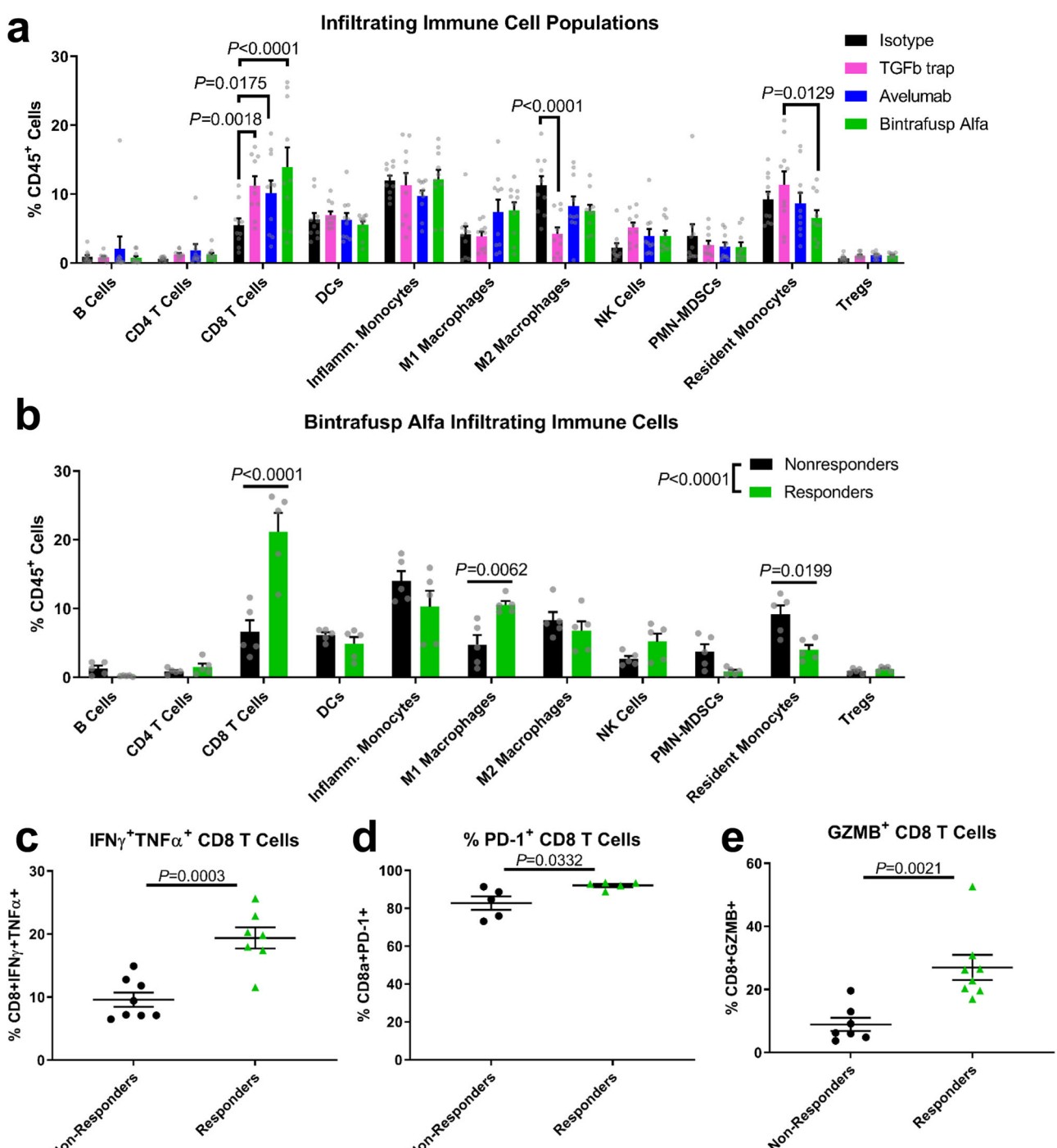

**Fig. 4 Responders to bintrafusp alfa have a more immune-permissive microenvironment and increased T cell activation when compared to nonresponders. a** Infiltrating immune cell populations of all mice treated with bintrafusp alfa (green bars, $n = 10$), its isotype control (black bars, $n = 10$), anti-PD-L1 (blue bars, $n = 10$), or TGFβ trap (pink bars, $n = 10$) after mass cytometric analysis and clustering. **b** Infiltrating immune cell populations of responder (green bars, $n = 5$) and nonresponder (black bars, $n = 5$) subpopulations of bintrafusp alfa-treated mice, with population differences determined as in (**a**) ($P < 0.0001$, $F = 11.17$, df = 10), with $P$-values for significant multiple comparisons as shown. **c** PD-1 expression on CD8 T cells from (**b**) as quantified by mass cytometry ($n = 5$). Tumor digests from mice treated as in (**a**) were treated for 4 h with brefeldin-A, ionomycin, and PMA and the proportion of **d** CD8$^+$ IFNα$^+$TNFα$^+$ and **e** CD8$^+$ Granzyme B$^+$ cells are shown ($n = 7$ nonresponders in black, $n = 8$ responders in green). Population differences for (**a**) ($P < 0.0001$, $F = 2.617$, df = 30) and (**b**) ($P < 0.0001$, $F = 11.17$, df = 10) were determined by two-way ANOVA and Sidak's multiple comparisons test with significantly altered populations as shown. Unpaired two-tailed $t$ tests were performed for (**c**–**e**), and all error bars represent the SEM.

IFNγ response. IFNγ is a potent promoter of antitumor immunity, inducing M1 macrophage polarization, increased T cell activation, and tumor cell apoptosis[40]. Furthermore, an IFNγ response can activate antigen-presenting cells and increase

CXCL9 secretion by dendritic cells and macrophages[41], leading to increased T cell activity within the TME.

Concurrent with that IFNγ response, we found increased CXCL9 expression in macrophages, dendritic cells, and

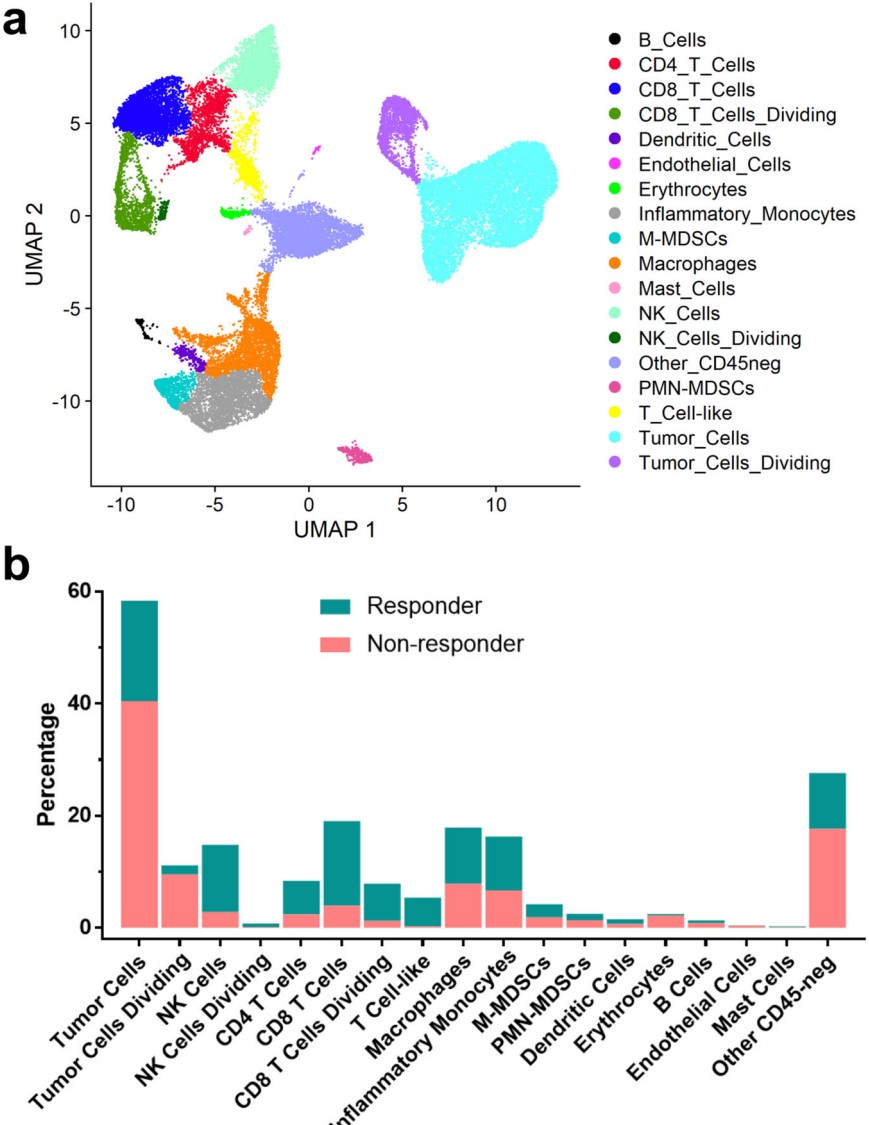

**Fig. 5 Single-cell RNA sequencing shows that bintrafusp alfa responders have increased infiltrating NK and T cells and decreased tumor cells compared to nonresponders. a** Combined UMAP clusters of cells from A223 tumors in bintrafusp alfa responders ($n = 2$) and nonresponders ($n = 2$) after cluster phenotyping. **b** Each cluster from (**a**) represented as a proportion of total responder cells (blue) and nonresponder cells (red).

inflammatory monocytes in responders (Fig. 6c). CXCL10, which is a part of the same signaling axis, also had increased expression in responders (Fig. 6c). CXCL9 and CXCL10 are notably context-dependent; they have both an autocrine tumorigenic effect on tumor cells and a paracrine antitumor effect on infiltrating lymphocytes and macrophages depending which cells are expressing its receptor, CXCR3[42]. Because CXCR3 expression was primarily in responder T and NK cells and absent from tumor cells (Fig. 6c), it appears to have an antitumor effect in this context and thus would induce T and NK cell activation. In addition, CCL5 is also responsible for the maintenance of tumor cells within the TME[43,44]. However, in combination with CXCL9 it promotes T cell infiltration into the tumor and allows an antitumor immune response in solid tumors[41]. Although responder tumors had elevated CCL5 expression across most phenotype categories, its co-expression with CXCL9 in responders indicates that it is acting to promote antitumor immunity. In summary, our single-cell RNA sequencing data indicates that the immune microenvironment of bintrafusp alfa responders promotes TME conducive to T cell activation and antitumor function.

**Anti-CXCR3 abrogates the response to bintrafusp alfa in A223 tumors, resulting in tumors indistinguishable from nonresponders to bintrafusp alfa**. To validate our single-cell sequencing results, we focused on the upregulation of CXCL9 and CXCL10 in bintrafusp alfa responders. To test whether this was required for a therapeutic response to bintrafusp alfa or just an indicator of responders, we treated mice with a blockading antibody against CXCR3, the receptor for CXCL9 and CXCL10, in the context of bintrafusp alfa therapy. We found that mice treated with bintrafusp alfa and anti-CXCR3 had significantly worse survival ($P = 0.0006$, $n = 10, 13$) and increased tumor volume ($P < 0.0001$ on day 27, $n = 10, 13$) relative to mice treated with bintrafusp alfa alone (Fig. 7a, b). In addition, treatment with anti-CXCR3 in combination with bintrafusp alfa was indistinguishable from either the isotype controls (survival $P = 0.3254$; day 27 tumor size $P = 0.4585$), or anti-CXCR3 alone (survival $P = 0.7416$; day 27 tumor size $P = 0.0723$, $n = 10, 13$) (Fig. 7a, b), suggesting that anti-CXCR3 prevents any antitumor effect of bintrafusp alfa.

We then analyzed the tumor infiltrating immune cell populations in mice treated as above and compared them to

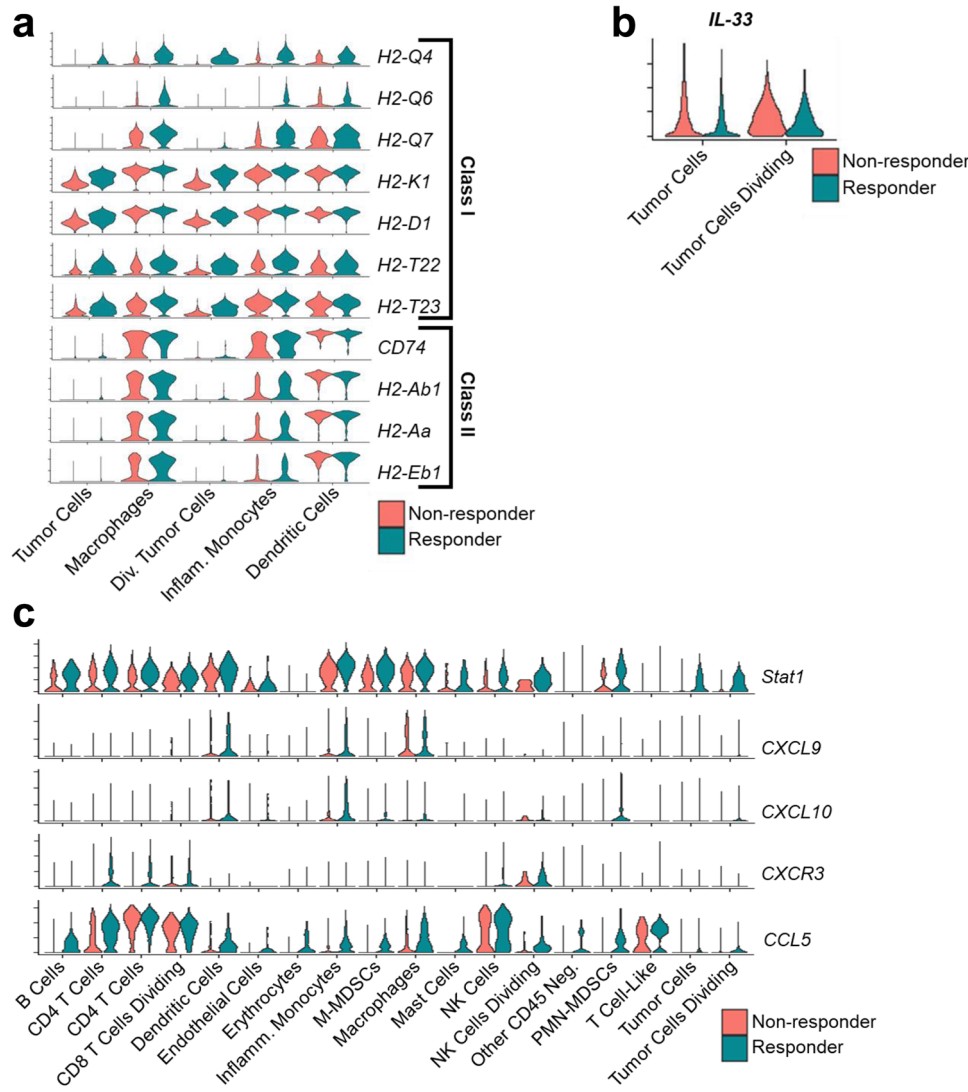

**Fig. 6 Bintrafusp alfa responders have elevated MHC-I and MHC-II expression, and chemokine expression consistent with antitumor immunity. a** Violin plots showing MHC-I and MHC-II subunit expression in the indicated UMAP clusters in A223 tumors responding (blue) and not responding (red) to bintrafusp alfa treatment. **b** Violin plots showing IL-33 expression in both A223 tumor cells and dividing tumor cells, separated by their responder (blue: responder, red: nonresponder) status. **c** Violin plots showing Stat1, CXCL9, CXCL10, CXCR3, and CCL5 expression in responder (blue) and nonresponder (red) tumors across all UMAP clusters. Violin plots show gene expression vertically on the y axis, and the binned cell count as the width on the x-axis.

bintrafusp alfa responders and nonresponders as defined previously. We found that mice treated with anti-CXCR3 in combination with bintrafusp alfa were indistinguishable from nonresponders to bintrafusp alfa ($P = 0.1333$ for resident monocytes, $P = 0.5340$ for inflammatory monocytes, and $P > 0.9999$ for all other immune cell populations), and had significantly fewer infiltrating CD8 T cells ($P < 0.0001$, $n = 5$) and M1 macrophages ($P = 0.0090$, $n = 5$) than bintrafusp alfa responders (Fig. 7c). These data indicate that CXCL9 and CXCL10 activity via CXCR3 promote CD8 T cell infiltration and M1 macrophage differentiation and infiltration in responders to bintrafusp alfa, and suggests a mechanism for why CXCR3 activity is required for the therapeutic effect of bintrafusp alfa.

**Discussion**
We used multiple SCC tumor models and therapeutics to show that when TGFβ inhibition is combined with PD-L1 blockade, the effect on survival is improved over either single agent alone, with responding mice exhibiting complete tumor eradication and long-term antitumor immunity. Although we show a link

between TGFβ signaling and tumor cell PD-L1 expression in vitro, our current study reveals that PD-L1 expression in the TME appears to be more complex and is regulated independently of the presence of TGFβ. Our Smad4-mutant tumor model lacks PD-L1 expression in vitro, and although the in vivo tumors were PD-L1-positive, the PD-L1 expression was primarily on infiltrating immune cells rather than on the tumor cells themselves. Recent studies have shown that even when patients have a PD-L1-negative biopsy, myeloid PD-L1 expression still functions as a predictor of improved prognosis[45]. Furthermore, stromal PD-L1 expression served as a better predictor of prognosis than tumor PD-L1 in breast cancer[46]. Here, we have presented an in vivo model of SCC that exhibits a similar predictive response from myeloid PD-L1 expression despite a lack of PD-L1 expression on tumor cells per se. This carries important clinical implications, because it highlights the importance of quantifying whole-tumor PD-L1 in a biopsy instead of focusing solely on tumor epithelial cells; current FDA guidelines for the administration of PD-1 blockade in HNSCC reflect this, using a PD-L1 scoring system that includes stromal PD-L1 expression[47], although no similar

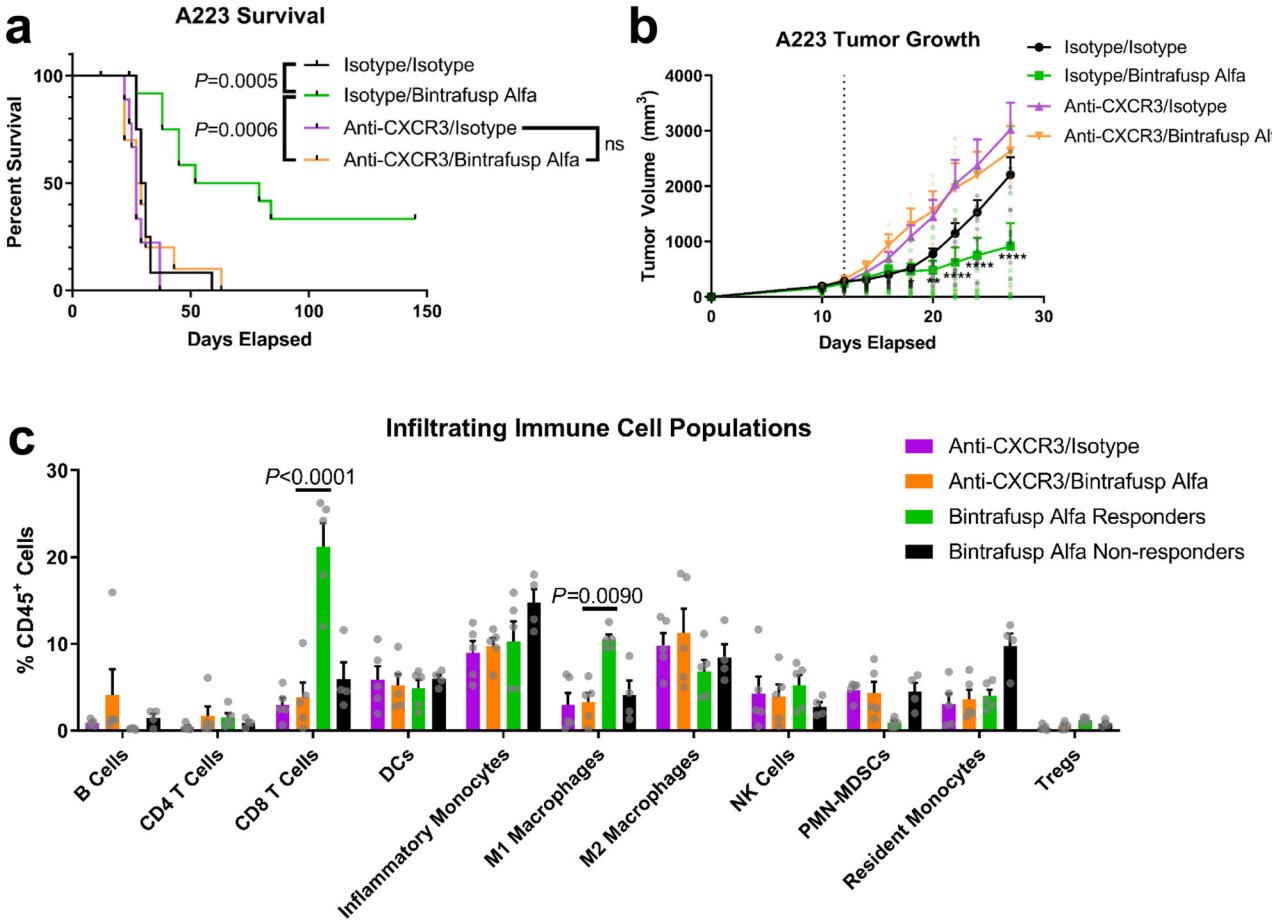

**Fig. 7 Anti-CXCR3 abrogates the response to bintrafusp alfa in A223 tumors, resulting in tumors indistinguishable from nonresponders to bintrafusp alfa. a** Survival post-injection of C57BL/6 mice bearing A223 tumors that were treated with double isotype controls (black line, $n = 12$), isotype and bintrafusp alfa (green line, $n = 13$), anti-CXCR3 and isotype control (purple line, $n = 9$), or anti-CXCR3 and bintrafusp alfa (orange line, $n = 10$). **b** Tumor growth in mice treated from (**a**). Survival comparisons were performed using the log-rank Mantel–Cox test. Differences between isotype/bintrafusp alfa (green) and anti-CXCR3/bintrafusp alfa (orange) in (**b**) were identified by two-way ANOVA ($P < 0.0001$, $F = 3.848$, df = 27) with multiple comparisons corrected by Tukey's method and indicated by * for $P = 0.0328$, ** for $P = 0.0028$, and **** for $P < 0.0001$. **c** Infiltrating immune cell populations of mice treated with anti-CXCR3 in combination with bintrafusp alfa (orange bars, $n = 5$) or its isotype control (purple bars, $n = 5$); mice were also treated with bintrafusp alfa alone and separated into nonresponders (black bars, $n = 4$) and responders (green bars, $n = 5$). Total population differences were determined by two-way ANOVA ($P < 0.0001$, $F = 5.979$, df = 30) and Sidak's multiple comparisons test with significantly altered populations between bintrafusp alfa responders (green) and anti-CXCR3 in combination with bintrafusp alfa (orange) are shown. Survival comparisons in (**a**) were performed using the log-rank Mantel–Cox test, and all error bars represent the SEM.

guidelines exist for other SCCs. Additionally, high local concentrations of PD-L1 or TGFβ in the TME may allow a bifunctional fusion protein like bintrafusp alfa to co-localize simultaneous TGFβ and PD-L1 inhibition in the TME, rather than systemic distribution of a small molecule inhibitor such as LY2109761. Indeed, a study on humanized mice treated with bintrafusp alfa had durable antibody concentration within the TME and increased tumor TGFβ depletion relative to the periphery[48].

While bintrafusp alfa increased CD8 T cell infiltration into A223 tumors, it did not significantly increase their TNFα and IFNγ production, initially suggesting that T cell activation was still impeded. However, after separating groups into responders and nonresponders to bintrafusp alfa, it became clear that responders had a more permissive TME for effector T cell function. This differentiation between bintrafusp alfa responders and nonresponders highlights the importance of parsing statistically insignificant results; although there were no statistical differences in immune cell infiltrates between treatment groups, that was due to the bintrafusp alfa-treated mice consisting of two

distinct outcomes. Notably, we found that this is not simply an effect of tumor size; only responders and nonresponders exhibited changes in infiltrating immune cell populations, while the isotype and single agents produced no differences between small or large tumor infiltrates. Previous research has shown that as tumors increase in size, their immune microenvironment changes to be generally less immunogenic[49]. Because we only saw that effect in bintrafusp alfa-treated tumor recipients, it appears that the responder categorization is not simply an effect of tumor size, but rather a distinct tumor profile that was masked by nonresponding mice. Responders and nonresponders were identified four days after treatment initiation, indicating that the changes we observed in the A223 TME are early effectors of the response to bintrafusp alfa. All tumors in survival experiments either rapidly reached endpoints or were completely eradicated, indicating that the differentiation of these two subpopulations is reflected in long-term tumor growth data as well. Future studies will focus on identifying whether there are underlying predictive traits in tumor-bearing mice that can determine whether they will respond to bintrafusp alfa treatment.

Although numerous infiltrating immune cell populations were altered to be more immune-permissive in responders, we showed that the effectiveness of bintrafusp alfa ultimately requires CD8$^+$ T cells; this confirms previous preclinical research that found that both NK and CD8$^+$ T cells contributed to the effectiveness of bintrafusp alfa[22]. The A223 tumor immune landscape was dramatically altered to permit effector T cell function, resulting in a T cell-mediated antitumor response that durably maintained antitumor immunity. This suggests that effector T cells against tumor-specific antigens were generated as a part of the tumor clearance. Interestingly, single-cell RNA sequencing identified slightly different populations of infiltrating immune cells; NK cells and CD4 T cells were notably more indicative of bintrafusp alfa responders in that context than by mass cytometry analysis. However, cells used for CytoF analysis are fixed and permeabilized, while live unfixed cells are used for scRNAseq with an additional live/dead cell selection step, which may result in different biases from each analysis method. Thus, single-cell sequencing and mass cytometry are most effectively used separately to identify changes in cell populations as a result of treatment, rather than as an attempt to empirically quantify infiltrating cell populations. Additionally, the presence of responders and nonresponders highlights the importance of personalized immunotherapy; although all the mouse recipients were genetically identical, there were two distinct outcomes of mice treated with bintrafusp alfa. This may be due to clonal differences in tumor-initiating cells, or it may be caused by host T cell clonotype variation between mice[50]. Although responders had fewer tumor cells than nonresponders, the reduction was evenly spread across all UMAP clusters that made up the "tumor cells" and "dividing tumor cells" categories; this suggests that tumor cell heterogeneity alone is not an indicator of responders to bintrafusp alfa therapy. Identifying the differences between these groups is just the first step toward being able to use the tumor landscape to predict whether a patient will be a responder or a nonresponder, and future experiments will focus on identifying changes in T cell receptor diversity as a predictor of responders to bintrafusp alfa.

Single-cell gene expression analysis has identified the importance of the context of chemokine presence in the TME; a whole-tumor analysis would not reveal the difference between paracrine and autocrine effects of context-dependent chemokines. Circulating CXCL9 is a predictor of poor prognosis in HNSCC;[51] however, in the context of CXCR3 expression in bintrafusp alfa responder tumors, it has the potential to induce antitumor immunity rather than promote tumor expansion. We confirmed this by finding that anti-CXCR3 blocked the therapeutic effect bintrafusp alfa, resulting in tumor growth, survival, and a TME indistinguishable from nonresponders to bintrafusp alfa. Because CXCR3 activity is required for tumor eradication in this model, the prognostic effect CXCL9 in HNSCC previously reported should be evaluated based on the context of where CXCR3 is expressed; CXCL9- or CXCL10-positive tumors with CXCR3 expressed on infiltrating immune cells may be indicative of a therapeutic response. Future studies will address whether CXCR3 expression on tumor cells results in a more tumorigenic TME in the context of CXCL9 or CXCL10 secretion.

The expression of cytokines and chemokines that induce the activation and migration of pro-inflammatory myeloid cells could also explain the shift in their populations in the TME that we observed by mass cytometry. This suggests that even if effector T cells infiltrated the microenvironment of nonresponders, their function would be inhibited by the other cells within the tumor; any immunotherapy in that environment would need to first induce a change in the topography of the immune microenvironment as a whole before it would release effector T cells

from immune checkpoint inhibition. The list of genes that identify responders (CXCL9, CXCL10, CCL5, Stat1, and MHC subunit genes) is notably similar to IFNγ-mediated predictors of response to PD-1 blockade in melanoma[52]. Here, we have further elaborated on that to identify early alterations in the TME landscape that are either responsible for or responding to those signaling changes in SCC tumors responding to bintrafusp alfa. These have the potential to be used to rapidly identify patients with the capacity to respond to combined TGFβ and PD-L1 blockade in the clinic.

In summary, we have demonstrated the effectiveness of simultaneous blockade of TGFβ and PD-L1 in mouse SCC models, targeting both TGFβ-secreting tumor cells and PD-L1$^+$ myeloid cells within the tumor (Fig. 8a, b). We identified changes in the immune microenvironment that allow mice to respond to therapy and clear the tumor in contrast to the immune-suppressive TME of nonresponders (Fig. 8c). These results suggest that evaluating the PD-L1 expression in myeloid cells of human SCC patients may help predict immunotherapy responders and demonstrate that combined TGFβ inhibition and PD-L1 blockade is an effective treatment combination that can be targeted in patients with "immune desert" SCC. Bintrafusp alfa is currently in Phase II clinical trials and the research presented here demonstrates its potential effectiveness in SCC and other tumor types commonly associated with TGFβ dysregulation. Future studies will use our single-cell sequencing results to identify biomarkers that predict a response to combined TGFβ inhibition and PD-L1 blockade, and to develop more mouse models of aggressive SCC that can be used to identify factors in the immune landscape that contribute to the success or failure of immunotherapy.

## Methods

**Cell lines.** The A223 cell line was derived from K15-CreΔPR1. Smad4$^{-/-}$Kras$^{G12D}$ mice as previously described[31], and then passaged in syngeneic C57BL/6 mice (Jax Laboratories) twice before isolating from a tumor-draining lymph node metastasis. A223 cells were cultured in DMEM complete media (10% fetal bovine serum (FBS) and 1% penicillin). LY2 cells have been previously described[33] and were cultured in DMEM/F12 complete media (10% FBS and 1% penicillin). Primary mouse normal keratinocytes were generated from minced tongue tissue and cultured in Keratinocyte Serum-Free Media (ThermoFisher) to differentiate them from other cells[53]. A1419 SCC cells were derived from spontaneous 4-NQO-driven tongue carcinomas following published protocols (manuscript in preparation)[54].

**Mice and in vivo cell injections.** Female C57BL/6 mice (Jackson Laboratories), athymic nude mice (Charles River), mTmG mice in a C57BL/6 background (ROSA$^{mT/mG}$, Jackson Laboratories), and CD8-knockout mice (B6.129S2-Cd8a$^{tm1Mak}$/J, Jackson Laboratories) between 6 and 8 weeks old were injected with 100,000 A223 cells suspended in PBS and 50% Matrigel Basement Membrane Matrix (Corning) to a final volume of 50 µl subcutaneously on their flank. Rechallenge experiments used the same injection protocol on the opposite flank. Treatment began either at tumor injection (Fig. 1c, d) or when tumors were established and reached a size of 100–200 mm$^3$ after approximately 13 days (all other experiments). Because LY2 cells were derived from an oral SCC, 1,000,000 LY2 cells were injected bucally into Balb/c mice (Jax Laboratories) in 50% Matrigel Basement Membrane Matrix (Corning) to a final volume of 50 µl, and treatment was started when tumors became palpable approximately 5 days post-injection. All mice were maintained under pathogen-free conditions in the vivarium facility of University of Colorado AMC. Tumor length and width were measured with calipers, and prolate ellipsoid tumor volume was calculated as $\pi(\text{length} \times \text{width}^2)/6$. Animal work was approved by the Institutional Animal Care and Use Committee of the University of Colorado, Anschutz Medical Campus (Aurora, CO).

**In vivo mouse treatments.** Mice were treated with LY2109761 (Ely Lilly, 75 mg/kg p.o., daily for 2 weeks) suspended in carboxymethylcellulose as previously described[55], bintrafusp alfa (EMD Serono, 492 µg/mouse IP, 3 per week for 2 weeks), anti-PD-L1 (Avelumab, EMD Serono, 400 µg/mouse IP, 3 per week for 2 weeks), bintrafusp alfa with a mutated anti-PD-L1 moiety as a TGFβ trap (EMD Serono, 492 µg/mouse IP, 3 per week for 2 weeks), or anti-PD-L1 with a mutated PD-L1 binding site as an isotype control (EMD Serono, 400 µg/mouse IP, 3 per week for 2 weeks). For combination treatment with LY2109761, mice were treated with either anti-PD-L1 (BioXCell, 250 µg/mouse IP, 3 per week for 2 weeks) or an

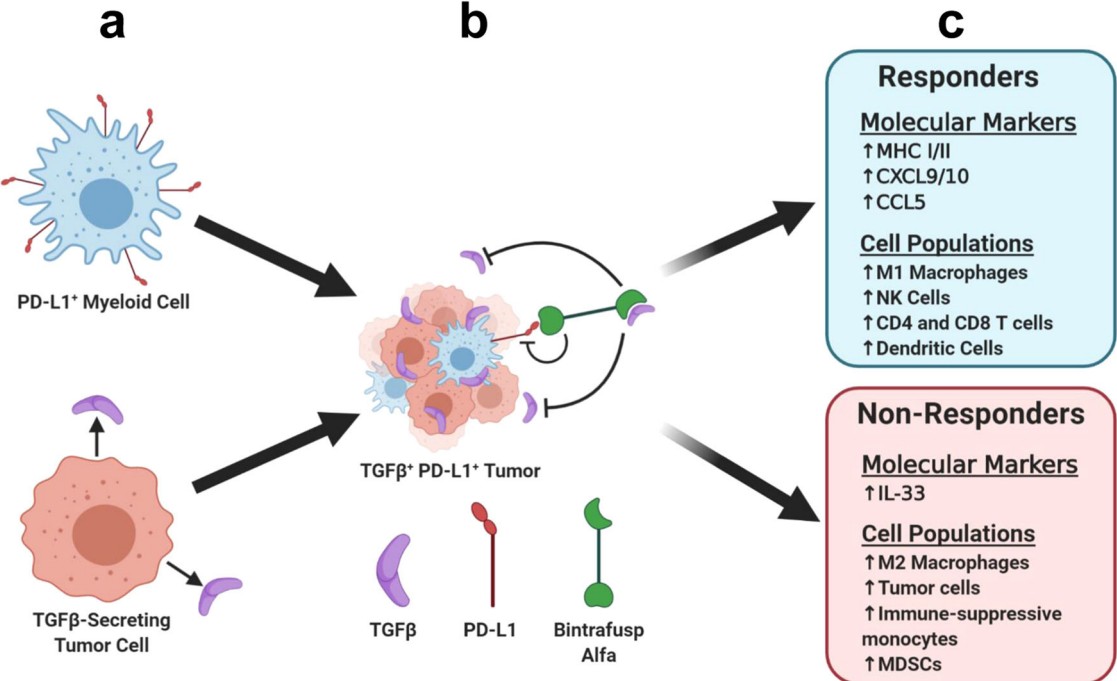

**Fig. 8 Bintrafusp alfa targets TGFβ and PD-L1 to generate distinct therapeutic responders and nonresponders. a** TGFβ-secreting tumor cells and PD-L1+ myeloid cells combine to form **b** an immune-suppressive TME, which can be targeted by bintrafusp alfa. **c** Distinct profiles of bintrafusp alfa responders and nonresponders emerge after therapy.

isotype control (BioXCell, 250 μg/mouse IP, 3 per week for 2 weeks). For combination treatments with anti-CXCR3, mice were additionally treated with anti-CXCR3 (BioXCell, 200 μg/mouse) or its isotype control (BioXCell, 200 μg/mouse) every other day starting one day before bintrafusp alfa treatment. All antibodies diluted with PBS to a 100 μl final volume. Survival studies were conducted until mice reached endpoints of 20% weight loss, severe tumor ulceration, or tumors reaching 2 cm in diameter for subcutaneous tumors or 1.5 cm for oral tumors. Diagnostic experiments were performed on tumors after 5 days of treatment (2 days after the 2nd dose) when tumor regression or progression could be differentiated.

**TGFβ ELISA.** 200,000 mouse normal keratinocytes and A223, LY2, A1419 cells were plated on a 6-well plate for 24 h to adhere. Media was replaced with 1.5 mL of DMEM/F12 supplemented with 0.5% FBS, which was collected after 24 h for analysis. Media samples were then analyzed for TGFβ levels using the Mouse/Rat/Porcine/Canine TGF-β1 Quantikine ELISA kit (R&D Systems) according to the manufacturer's instructions, corrected for baseline TGFβ levels from the media alone.

**Statistics and reproducibility.** Single comparisons were performed using unpaired two-tailed t tests, with corrections for multiple comparisons performed using the Holm–Sidak method. Multiple comparisons were performed either using a two-way ANOVA with post-hoc analyses using Sidak's multiple comparisons test, or by one-way ANOVA with post hoc analyses using Dunnett's multiple comparisons test. Survival comparisons were performed using the log-rank Mantel–Cox test. Error bars represent the mean ± the standard error of the mean of biologically independent replicates. In vitro experiments (Figs. 1a, 2a, and Supplementary Fig. 1a, b) are representative of three independent experiments. Figure 1c, d is a combination of three independent experiments, and Fig. 1e–h, Fig. 3a–f, h, and Supplementary Fig. 2a are representative of three independent experiments. Figure 4 and Supplementary Figs. 4 and 5 are a combination of two independent experiments that are representative of four total independent replicates. Single-cell RNA sequencing data is presented as the combination of two independent experiments (1 responder and 1 nonresponder each).

**pSmad3 IHC and quantification.** Formalin-fixed paraffin-embedded A223 tumor samples were stained using anti-phospho-Smad3 (clone EP823Y, Abcam cat. ab52903) diluted 1:250, using the Pierce Peroxidase IHC Detection Kit (Thermo-Fisher), and using citrate pH 6.0 for antigen unmasking. Slides were scanned with an Aperio AT2 scanner (Leica), and 5 images per slide were collected with the Aperio software. Images were then processed using QuPath v0.2.3[56], selecting positive cells using default settings for DAB chromagen with a minimum threshold

of 0.2479. The percentage of positive cells was averaged for all 5 images per sample, with the averages for each sample ($n = 4$) reported in the text.

**Multispectral fluorescence imaging IHC.** Twenty-three formalin-fixed paraffin-embedded human HNSCC samples were stained using the Opal 7-Color Automation IHC Kit-50 Slide according to manufacturer's instructions (Akoya Biosciences) using the following antibodies and antigen retrieval processes: with antibodies against PD-L1 (clone E1L3N, Cell Signaling Technologies cat. 13684S, Opal 620), CD11c (clone 5D11, Sigma Cell Marque cat. 111M-15, Opal 650), CD68 (clone KP1, Dako cat. M0814, Opal 690), and pan-CK (clone AE1/AE3, Dako cat. M351501-2, Opal 540) on a Bond RX autostainer (Leica Biosystems). Slides were dewaxed (Leica), heat treated in ER2 or ER1 antigen retrieval buffer depending on the antibody for 20 min at 93 °C (Leica), blocked in Antibody (Ab) Diluent (Akoya Biosciences), incubated for 30 min with the primary Ab, 10 min with horseradish peroxidase (HRP)-conjugated secondary polymer (anti-rabbit and anti-mouse, Akoya Biosciences), and 10 min with HRP-reactive OPAL fluorescent reagents (Akoya Biosciences). Slides were washed between staining steps with Bond Wash (Leica) and stripped between each round of staining with heat treatment in antigen retrieval buffer. After the final heat treatment in antigen retrieval buffer, the slides were stained with spectral DAPI (Akoya Biosciences), and coverslipped with Prolong Diamond mounting media (Thermo Fisher).

**Multispectral fluorescence imaging and analysis.** Whole slide scans were imaged on the Vectra 3.0 Automated Quantitative Pathology Imaging System (Akoya) using the 20× objective. We randomly selected 12 multispectral image regions each slide using Phenochart software (Akoya Biosciences) at 40× magnification, which were then analyzed with inForm software (v2.4.10, Akoya) to unmix adjacent fluorochromes, subtract autofluorescence, segment the tissue into tumor and stroma regions, segment the cells into nuclear, cytoplasmic, and membrane compartments, and to phenotype the cells according to morphology and cell marker expression based on either their PD-L1 status or their expression of CD68, CD11c, and CK (CD68−CD11c−CK− cells were defined as "other"). Independent projects were created to phenotype each cellular marker, then merged, consolidated, and analyzed in R Studio using the phenoptrReports plug-in (Akoya Biosciences) to quantify the total number of PD-L1+ cells that overlapped CD68/CD11c/CK/other cells. We also used phenoptrReports to quantify entire-cell PD-L1 expression in each phenotype category in all 23 samples. Use of human samples was approved by IRB of University of Colorado AMC (COMIRB Protocol 16-2436).

**Single-cell tumor digestion.** Tumors were excised from mice, minced, and digested with Liberase TL (Roche, 0.13 Weunsch units/ml) in GentleMACS C-tubes (Miltenyi) following the GentleMACS user protocol (Miltenyi). Digests

were ground through a 100 μm strainer, red blood cells lysed with eBioscience 1× red blood cells Lysis Buffer (Invitrogen) for 5 min, and finally strained through a 70 μm strainer before counting and downstream analyses.

**mTmG flow cytometry staining and analysis**. Totally, 1.5 million cells from tumors in mTmG mice were stained following the BD Cytofix/Cytoperm Fixation/Permeabilization microplate staining protocol (BD Biosciences), using the following antibodies: CD16/32 Fc Block (clone 2.4G2, BD Biosciences cat. 553142, 1:100), CD45 (clone 30-F11, BioLegend cat. 103127, 1:350), PD-L1 (clone 10F.9G2, BioLegend cat. 124313, 1:100), CD11b (clone M1/70, BioLegend cat. 101223, 1:100), CD11c (clone N418, BioLegend cat. 117323, 1:100). Cell lines were stained with PD-L1 (clone 10F.9G2, BioLegend cat. 124307, 1:100). Dead cells were excluded using Ghost Dye Violet 510 Dead Cell Stain Kit (Tonbo Biosciences) according to manufacturer protocol, and compensation was calibrated with single-color controls on Ultracomp eBeads (Invitrogen). Cells were quantified using a Gallios 561 cytometer (Beckman Coulter) and analyzed using Kaluza analysis software (Beckman Coulter) with fluorescence-minus-one controls to verify gating.

**Smad4 siRNA and in vitro PD-L1 and Smad4 protein expression analysis**. Totally, 400,000 LY2 cells were plated on a 6-well plate and allowed to reach 50% confluence. Cells were transfected with either Qiagen GeneSolution siRNAs against Smad4 (Qiagen cats. SI01426215, SI01426222, and SI01426229) or Qiagen AllStars Negative Control siRNA (Qiagen cat. 1027280) using Lipofectamine RNAiMax (Invitrogen) according to manufacturer's instructions. siRNA transfection complex consisting of 1.67 μl 20 μM siRNA, 2 μl Lipofectamine, and 163 μl OptiMEM were added to 1.5 mL of cell media and left for 48 h before cell trypsinization and either flow cytometry or western blot analysis. Cells were stained for flow cytometry using the BD Cytofix/Cytoperm Fixation/Permeabilization microplate staining protocol (BD Biosciences), using the following antibodies: CD16/32 Fc Block (clone 2.4G2, BD Biosciences cat. 553142, 1:100) and PD-L1 (clone 10F.9G2, BioLegend cat. 124307, 1:100). PD-L1 specificity was confirmed using 10 ng/ml IFNγ as a positive control and mouse IgG2b (BioLegend cat. 400311) as an isotype control. Cells were counted using a Gallios cytometer (Beckman Coulter) and analyzed using Kaluza analysis software (Beckman Coulter) using the isotype control to set 0% PD-L1 positivity.

For western blot analysis, cells treated with Smad4 siRNA as above were lysed with Lysis Buffer M (Roche) supplemented with cOmplete mini (Roche), run on a 4–15% tris-glycine gradient gel (BioRad), transferred to nitrocellulose membrane, stained with antibodies against Smad4 (AbCam cat. Ab40759, clone EP618Y) and actin (Santa Cruz BioTech cat. SC-47778, clone C4) following the Li-Cor Near-Infrared Western Blot Detection protocol (Li-Cor) and using goat anti-rabbit (Rockland Inc. cat. 611-145-002) and donkey anti-mouse (Li-Cor cat. 926-68072) secondary antibodies. Blots were imaged on an Odyssey 9120 Digital Imaging System (Li-Cor).

**Mass cytometry staining and analysis**. Three million cells were stained with Maxpar antibodies listed in Supplementary Table 1 according to manufacturer protocol for intracellular staining (Fluidigm), with the addition of Cell-ID 20-Plex Pd Barcoding Kit (Fluidigm) barcoding after cell surface antibody cocktail staining. Cells were analyzed on a Helios mass cytometer (Fluidigm), and normalized FCS files were debarcoded with Debarcoder v1.4 (Fluidigm) before being uploaded to Cytobank (Cytobank, Inc.) for statistical analysis. Dead cells were excluded based on positive cisplatin staining, and nucleated cells were confirmed with IR-Dye (Fluidigm). Live, nucleated, CD45+ cells were then clustered using the viSNE clustering protocol (5000 iterations, perplexity of 30, and a theta value of 0.5), and gated based on individual marker expression. Population trees were generated based on the percent of total CD45+ cells for each tumor.

**Ex vivo T cell stimulation and flow cytometry**. Single-cell suspensions from tumors were plated in a 12-well plate (2,500,000 cells per sample per well) and stimulated with 25 ng/mL phorbol 12-myristate 13-acetate (PMA) (Sigma Aldrich), 485 ng/mL ionomycin (Sigma Aldrich), 5 μg/mL BFA (Biolegend), 10% FBS, 1× Antibiotic–Antimycotic (Gibco), and 100 μM β-mercaptoethanol in RPMI media for 4 h at 37 °C. Stimulated cells were removed from the plate, washed, and stained with the following stains and surface antibodies: Live/Dead Aqua (Invitrogen), CD8 (clone 53-6.7, BioLegend cat. 100747, 1:250), CD45 (clone 30-F11, BD Bioscience cat. 564279, 1:350), and CD4 (clone RM4-5, BioLegend cat. 100563, 1:250). After surface staining, cells were fixed and permeabilized with the BD Fixation/Permeabilization kit and stained with intracellular antibodies: IFN-γ (clone XMG1.2, eBioscience cat. 12-7311-41, 1:100), TNFα (clone MP6-XT22, BioLegend cat. 506307, 1:100), and granzyme B (clone QA16A02, BioLegend cat. 372203, 1:100). Cells were analyzed using a BD LSRFortessa X-20 cytometer and data were analyzed by FlowJo software.

**Single-cell RNA sequencing**. Single-cell suspensions from two bintrafusp alfa responders and two bintrafusp alfa non-responders were prepared as above, and after red blood cell lysis dead cells were eliminated using the Dead Cell Removal MACS kit (Miltenyi Biotec) and diluted to 1,000,000 cells/mL in PBS. Approximately, 10,000 cells per sample were captured and barcoded using the Single-Cell 3′

v3 kit on the Chromium Cell Capture Machine (10× Genomics) before sequencing on an Illumina NovaSEQ 6000 at the University of Colorado at Anschutz Genomics and Microarray Core to a target read depth of 100,000 reads per cell. Raw reads were debarcoded and mapped to the mm10 reference genome using the CellRanger count pipeline (10× Genomics), and then analyzed with the Seurat v3.1.5R package[57,58] as follows:

1. Cells with fewer than 200 features and greater than 30% mitochondrial RNA were excluded, resulting in 38,491 total cells from four samples being analyzed.
2. Responder and nonresponder metadata were added for each cell using AddMetaData.
3. Gene expression was normalized across all samples using NormalizeData, and the top 2000 variable features were identified with FindVariableFeatures and integrated with FindIntegrationAnchors and IntegrateData.
4. Integrated variable features were used to cluster and visualize all cells by UMAP with ScaleData and RunUMAP, FindNeighbors, and FindClusters.
5. Individual clusters were phenotyped by comparing their gene expression to the PanglaoDB single-cell RNA sequencing database of known cell categories[59], supplemented by a curated list of commonly known markers where indicated. A223 tumor cells were identified from gene expression markers found from bulk RNA sequencing of that cell line in vivo (not presented here). Clustered heat maps were created with the pheatmap R package.
6. Clusters were then renamed and visualized by their UMAP coordinates, and cell counts for each cluster were reported per their responder status.
7. Differential gene expression for responder status within each cluster was identified using the FindMarkers function and plotted using the VlnPlot function in Seurat.

**Reporting summary**. Further information on research design is available in the Nature Research Reporting Summary linked to this article.

## Data availability
The data that support the findings of this study are available from the corresponding authors upon request. Single-cell RNA sequencing data and metadata is available via NCBI GEO (accession GSE161370). Raw western blot images are available in Supplementary Fig. 17 and the source data for all figures is available in Supplementary Data 1.

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

## Acknowledgements

We thank Dr. James Costello for the use of the AMC-Hertz computing cluster, Dr. Yao Ke for assistance with IHC quantification, the Human Immune Monitoring Shared Resource within the University of Colorado Human Immunology and Immunotherapy Initiative for their expert assistance in analysis of MSI images, the University of Colorado Cancer Center Flow Cytometry Shared Resource for their assistance with flow panel design and analysis (supported by the Cancer Center Support Grant P30CA046934) and the RNA Biosciences Initiative at the University of Colorado for assistance with single-cell RNA sequencing analysis. Bintrafusp alfa and its controls were provided for research purposes by EMD Serono Research & Development Institute, Inc., Billerica, MA, USA; a business of Merck KGaA, Darmstadt, Germany. A.A.S. was supported by NIH pre-doctoral training grants T32CA174648, T32AR007411, and T32GM007635. A.A.S. and D.R. are supported by the Marsico Family Endowed Chair of Head and Neck Cancer Research and an anonymous philanthropic donor to head and neck cancer research. This work is supported by NIH R01 DE24371 to Xiao-Jing Wang and Antonio Jimeno, DE027329 and DE 028420 to Xiao-Jing Wang and Jing H. Wang. Xiao-Jing Wang is supported by VA merit award I01 BX003232. R.A.W. is supported by a NIH F31 fellowship (F31DE027854). Schematics were created with BioRender.com.

## Author contributions

A.A.S. designed and performed experiments, analyzed the data, performed the bioinformatics analysis, and co-wrote the paper. R.A.W. performed CD8KO mouse experiments, ex vivo CD8 T cell stimulation, and analyzed the single-cell RNAseq data. S.H. collected and

analyzed in vivo mouse experiment data, and C.D.Y. generated the A1419 cell line. S.D.K. provided the LY2 cell line. A.J. provided the human HNSCC samples. S.H. supported animal experiments. D.R. and Y.L. provided access to bintrafusp alfa and guidance on its dosage and applications in SCC. J.H.W. provided the CD8KO mouse strain and performed ex vivo CD8 T cell stimulation. X.J.W. designed experiments and co-wrote the paper.

## Competing interests

The authors declare the following competing interests: Y.L. is an employee of EMD Serono Research & Development Institute, Inc., Billerica, MA, USA; a business of Merck KGaA, Darmstadt, Germany. The remaining authors declare no competing interests.
