## [Peer Review File · Communications Biology]

Reviewers' Comments:

Reviewer #1:

Remarks to the Author:

In this study, Strait and colleagues explore the use of the emerging TGF β ligand trap and PD-1 inhibitor Bintrafusp alpha. The authors conduct a series of in vivo studies mainly using Bintrafusp alpha, then explore immune alterations in the immune microenvironment of mice that demonstrated a therapeutic response and those that did not.

Though timely and interesting, this work suffers from being extremely descriptive in nature. The authors do an admirable job describing the immune infiltrate of responders and non-responders; however, they do not in any way show that their findings have a causative role in the apparent failure of bintrafusp alpha in the non-responders. Hence, though this is likely to be the case, they have merely characterized the local immune environment of without offering new mechanistic insight or true evidence of causation.

The authors are also limited by a very small sample size for many experiments, which show an N of 2 or 3 (N=1 in Figure S8) despite there being 6 responders. This raises concerns of unintentional selection bias due to limited sample size, which can be rectified by corroborating these findings in all animals using additional techniques e.g. immunohistochemistry, flow cytometry, or at minimum qPCR. Specific comments and suggestions are outlined below.

Major Comments

Figure 1

-The relevance of Figure 1B to this study is unclear. The authors suggest that SMAD4-mutant SCC tumors have high levels of TGF β , yet show only an mRNA association that does not account for mutation. The authors should instead examine TGF β expression in SMAD4 Wild Type and SMAD4 mutated tumors from this group, which can easily be performed using cBioportal.

-Please verify the effectiveness of LY2109761 in vivo by immunohistochemistry or western blot for pSMAD2/3 or nuclearization of SMAD4

Figure 2

-Figures 2a and 2b should be confirmed using other means, namely western blotting as the negligible change in mRNA does not seem to reflect the more dramatic change in PD-L1 expression by flow cytometry. Additionally, as the authors subsequently claim that the immune infiltrate is in fact the dominant source of PD-L1, the relevance of these experiments using tumor cells is also unclear. Perhaps a better experiment would be to repeat using these experiments the leukocyte populations of interest.

Figure 3

-As previously, please verify the effectiveness of TGF β signal inhibition in vivo by immunohistochemistry or western blot for pSMAD2/3 or nuclearization of SMAD4

Figure 4

-While the increase in CD8 cells in responders compared to non-responders is both compelling and expected, the immune infiltrate should be quantified across all groups in the main figures. The inclusion of these proper controls would likely only strengthen the authors' conclusions. Though some of this data tumor is included in the supplement, their analysis does not appear as rigorous as that of

the bisulfran alpha group.

Additionally, many of these analyses seem to be based on differences such as "large" and "small" tumor size, which appear rather arbitrary and occasionally rely on an N of 1. Composite averages across all groups should be clearly displayed and directly compared to the bisulfran alpha group.

-Beyond the mere number of CD8 cells from these mice, the authors should also evaluate the expression of cytotoxic markers such as Perforin and GranzymeB in by these CD8 T-cells.

Figures 5 and 6

-Though well performed, this uses a relatively small sample size and again and does not include a comparison to control mice or those from other treatment groups.

Minor Comments

-Gene names should be written using standard criteria e.g SMAD4 for human, Smad4 for mouse, etc.

-Several important studies describing concomitant TGFβ and PD-1 signal inhibition are missing, including those that have explored the combination of this approach and chemotherapy to enhance antigen presentation.

Reviewer #2:

Remarks to the Author:

Strait et al. tested different squamous cell carcinoma (SCC) mouse cell lines for their ability to secrete TGF-beta and for the therapeutic effect of a TGF-beta inhibitor alone, or combined with anti-PD-L1 immunotherapy. Most of the methods are clearly described, though for some experiments the sample size is rather small for the statistical tests that are performed. For the majority of their work, they used a syngeneic mouse SCC cell line, the A223 cell line, which was previously derived by the authors from a genetically engineered mouse model that spontaneously develops oral and skin SCC. The authors show that this cell line produces high levels of TGF-beta and that tumor growth is inhibited when blocking this growth factor as monotherapy or in combination with anti-PD-L1, but only in mice with an intact adaptive immune system. Recent literature has shown that TGF-beta is a major factor that drives primary or secondary resistance to immunotherapy with anti-PD-(L)1 agents. While the authors show that A223 cells themselves express little PD-L1, the myeloid cells within the tumor microenvironment (TME) do express this immune checkpoint ligand. Interestingly, the therapeutic treatment with Bintrafusp alfa (an agent that simultaneously can block TGFbeta and PD-L1) results in mice that respond to the treatment (and are protected upon rechallenge) but also a group of mice that are unresponsive to the treatment. Such a heterogeneous response would also be something one would expect in a head and neck SCC patient population. They go on to show that responder and non-responder mice have differences in their immune infiltrate, HLA class I expression on tumor (and several immune cells), and chemokine profiles. This work provides important clues where to look for, with regards to biomarkers and patient selection, when designing clinical studies using dual TGF-beta/PD-L1 inhibition for head and neck SCC, and potentially also other SCC tumors.

I have a few points that require additional attention or explanation by the authors:

- For the mass cytometry data n=3 responders were compared to n=2 non-responders. Clearly the error bars for the non-responders are quite large. It would strengthen the data, and also the statistics, if a third non-responder case could be included for the mass spec analysis. I don't think a two-way ANOVA (or T-test) should be used on unbalanced data since with n=2 a normal distribution of the data can't be determined.

- In figure 4, no differences in the percentage of CD4+ cells (out of CD45+ cells) is observed between the responders and non-responders based on the CYTOF data, whereas based on the scRNAseq data

(shown in figure 5) statistically significant differences in CD4 TILs are observed. The authors should explain the difference in outcome between these two methods.

- As an explanation for their finding that in mice, which are genetically identical, treatment with bintrafusp alfa generates two distinct outcomes, the authors suggest that this may relate to (subtle) differences in T cell clonotype which arise due to random TCR rearrangement in the thymus.

o Do the authors have evidence, or could they generate such evidence, that there is perhaps a less diverse TCR profile in the non-responding mice compared to the responding mice?

o Also do the scRNAseq data on the tumor cells, apart from changes in HLA class I expression, provide insight on whether tumor cell heterogeneity is different between the groups?

o Is it possible that epigenetic or metabolism may play a role in these differences: was there a difference in the weight of the mice not responding, which may be related to the fact that they had a different caloric intake?

minor points:

- On p8 an incorrect p-value is mentioned in the text with regards to the inflammatory and resident monocyte infiltration ($p=0.272$), where in figure 4 a p-value of $p=0.0272$ is reported.

- Do I understand correctly that for the scRNAseq experimental setup 10,000 cells were captured for each sample and the in total 40,0000 cells were sequenced? Or were the samples hashed prior to capturing the single cells and were in total 10,000 cells (mix of all 4 samples) sequenced? This should be described in more detail in the methods. Also, after applying the exclusion criteria as mentioned, how many cells were eventually used (for responder and non-responder samples) for the final analysis?

Reviewed by: Dr. Rieneke van de Ven

Reviewer #3:

Remarks to the Author:

The authors show that immune microenvironment profiles of responder in TGF β /PD-L1 blockade-treated SCC. In general, I think this paper is interesting and has contributions. However, I still have some concerns, suggestions or questions, which I would like the author(s) to address or answer.

1. In figure 1a, you measured TGF β in mouse cell lines. Did you measure human cell lines? Also, were those cells (MNK, A223, A1419) with repeated measurement 2 times. So, I would like to repeat more one time.

2. In figure 1b, you checked TGF β 1 vs SMAD4 mRNA expression in human cell patient samples. Did you check it in mouse cell lines (A223 cells)?

3. In figure 1 e-f, tumor growth was inhibited in C57BL/6 mice not nude mice. So, you suggest that systemic TGF β inhibition improved the effectiveness of T cell mediated immunotherapy. If it is right, did you check in other mice (Balb/c or etc)?

4. In supplement figure 4, you injected LY2 cells in Balb/c mice. If you want to say other mice also affect tumor growth, I think you need to A1419 cells also. Why did you inject LY2 in Balb/c and inject A1419 in C57BL6?

Response to Reviewers

We are grateful for the thorough and constructive comments from the reviewers. To adequately address all of the reviewers' concerns, we have performed additional *in vivo* and *in vitro* experiments and extensively revised our manuscript. As a result, we hope reviewers will find a much strengthened manuscript. Changes in the revised manuscript are indicated for line numbers below per journal's instruction. A point-by-point response is provided below.

Reviewer #1

"The authors...do not in any way show that their findings have a causative role in the apparent failure of bisulfran alpha in the non-responders."

Response: We took this criticism into the heart to perform additional studies. Our scRNA analysis identified that CXCL9/10 elevation in immune cells is one of the top indicators of responders to bintrafusp alfa. To determine if this is required for therapeutic response, we used an antibody blockade for their receptor CXCR3 in the context of bintrafusp alfa treatment. We found that anti-CXCR3 abrogated the therapeutic response to bintrafusp alfa, and that mice treated with CXCR3 antibodies have a TME indistinguishable from non-responders. We have added these data and related discussion to the revised manuscript (see revised Fig. 7, results on lines 292-313, discussion on lines 386-398).

"The authors are also limited by a very small sample size for many experiments, which show an N of 2 or 3 (N=1 in Figure S8) despite there being 6 responders."

Response: We have repeated the experiments that originally had small sample sizes. Because some phenotypes were rare (non-responders to bintrafusp alfa and small anti-PD-L1-treated tumors, for example), we have pooled the results between two independent repeats for Fig. 4 and supplementary Figs. 8 and 9, resulting in $n=10$ for each treatment group and $n=5$ for each size or responder category. We also increased the sample size for *ex vivo* T cell activation to $n=7$. This allowed us to statistically quantify populations of infiltrating immune cells more robustly (see revised Fig. 4 and Supplementary Figs. 8 and 9, results on lines 181-231, discussion on lines 340-360).

"The authors suggest that SMAD4-mutant SCC tumors have high levels of TGFB, yet show only an mRNA association that does not account for mutation. The authors should instead examine TGFB expression in SMAD4 Wild Type and SMAD4 mutated tumors from this group, which can easily be performed using cBioportal."

Response: Following the reviewer's suggestion, we have re-analyzed our cBioportal data to reflect *SMAD4* mutant status (often chromosomally deleted instead of point mutations) and its relationship to *TGFB1* mRNA expression, as revised in Figure 1b. We show that patient tumors with *SMAD4* deletions (combined homozygous deletion and one copy loss) have significantly greater *TGFB1* mRNA expression than *SMAD4* diploid tumors ($P=0.0011$) (see revised Fig. 1b, results on lines 90-94).

"Please verify the effectiveness of LY2109761 in vivo by immunohistochemistry or western blot for pSMAD2/3 or nuclearization of SMAD4." (editor) "The actual TGF-β signal inhibition in figures 1 and 3 should be verified by examining SMAD2/3 activation"

Response: We have added quantification of pSmad3 immunohistochemistry staining to the revised manuscript demonstrating that LY2109761, bintrafusp alfa, and its TGFβ trap control inhibit TGFβ signaling in our experiments (see revised Supplementary Figs. 1b, 1c, and 5a, results on lines 96-99 and 158-160, and methods on lines 489-497).

“Figures 2a and 2b should be confirmed using other means, namely western blotting as the negligible change in mRNA does not seem to reflect the more dramatic change in PD-L1 expression by flow cytometry. Additionally, as the authors subsequently claim that the immune infiltrate is in fact the dominant source of PD-L1, the relevance of these experiments using tumor cells is also unclear. Perhaps a better experiment would be to repeat using these experiments the leukocyte populations of interest.”

Response: We agreed with the reviewer’s view that, the flow cytometry data more accurately represents the functional availability of PD-L1 on the cell surface, therefore we have replaced the qPCR data with the flow cytometry data (Fig. 2a). We have also clarified in the results section that the LY2 data is recapitulating previous research on the relationship between TGFβ signaling and PD-L1 expression as a positive control, and that A223 tumors represent a departure from that relationship both *in vitro* (see results on lines 118-128) and *in vivo* (see Fig. 2b, c, and lines 129-137).

“While the increase in CD8 cells in responders compared to non-responders is both compelling and expected, the immune infiltrate should be quantified across all groups in the main figures. The inclusion of these proper controls would likely only strengthen the authors’ conclusions. Though some of this data tumor is included in the supplement, their analysis does not appear as rigorous as that of the bintrafusp alfa group.”

Response: Following the reviewer’s suggestion, we have added immune infiltrates of all groups to Fig. 4. Due to space limitation, we have moved the comparisons between small and large tumor sizes to Supplementary Figs. 8 and 9 (See revised Fig. 4, Supplementary Figs. 8 and 9).

“Additionally, many of these analyses seem to be based on differences such as “large” and “small” tumor size, which appear rather arbitrary and occasionally rely on an N of 1. Composite averages across all groups should be clearly displayed and directly compared to the bintrafusp alfa group.”

Response: Because technical limitations of our mass cytometry analyzer prevent us from running more than 20 samples per experiment, we have pooled results from two independent repeats to increase sample size for our mass cytometry experiments resulting in a combined $n=10$ for each drug treatment group, and $n=5$ for each sub-population based on tumor size or responder status. We also presented composite averages in Supplementary Fig. 8e (See revised Fig. 4 and Supplementary Figs. 8 and 9, and results on lines 187-229).

“Beyond the mere number of CD8 cells from these mice, the authors should also evaluate the expression of cytotoxic markers such as Perforin and GranzymeB in by these CD8 T-cells.”

Response: We have added granzyme B to our previous analysis of TNFα and IFNγ as markers of CD8 T cell activation (Fig. 4c, results on lines 195-197 and 183-231).

(scRNAseq)“Though well performed, this uses a relatively small sample size and again and does not include a comparison to control mice or those from other treatment groups.”

Response: Because bintrafusp alfa treatment group had mixed responders and non-responders, comparing this group with other groups (which had very few or no responders) without stratifying mice into responders vs. non-responders resulted in too high variability to be distinguishable from one another, similar to our results in presented in Fig. 4. Further, in addition to a high cost of scRNAseq (\$7,000 per sample), coordinating our animal and lab access with two core services for live cell capturing and sequencing during COVID has been extremely difficult. Taken together, we chose to focus on analyzing what differentiates responders from non-responders to bintrafusp alfa, and subsequently validated one of

the top candidates (CXCL9/10 and their receptor CXCR3) across all treatment groups (see revised Fig. 7, results on lines 292-313, discussion on lines 386-398, and methods on lines 462-464).

“Gene names should be written using standard criteria e.g SMAD4 for human, Smad4 for mouse, etc.”

Response: We have corrected the genetic notations in the manuscript (see references to *Stat1* in Fig. 6c and on lines 272 and 406).

“Several important studies describing concomitant TGF β and PD-1 signal inhibition are missing, including those that have explored the combination of this approach and chemotherapy to enhance antigen presentation.”

Response: We have updated our references with additional related citations (see lines 54-59).

Reviewer #2

“For the mass cytometry data $n=3$ responders were compared to $n=2$ non-responders. Clearly the error bars for the non-responders are quite large. It would strengthen the data, and also the statistics, if a third non-responder case could be included for the mass spec analysis. I don’t think a two-way ANOVA (or T-test) should be used on unbalanced data since with $n=2$ a normal distribution of the data can’t be determined.”

Response: Because technical limitations of our mass cytometry analyzer prevent us from running more than 20 samples per experiment, we have pooled results from two independent repeats to increase sample size for our mass cytometry experiments resulting in a combined $n=10$ for each drug treatment group, and $n=5$ for each sub-population based on tumor size or responder status (See revised Fig. 4 and Supplementary Figs. 8 and 9, and results on lines 181-231).

“In figure 4, no differences in the percentage of CD4+ cells (out of CD45+ cells) is observed between the responders and non-responders based on the CYTOF data, whereas based on the scRNAseq data (shown in figure 5) statistically significant differences in CD4 TILs are observed. The authors should explain the difference in outcome between these two methods.”

Response: Cells used for CytoF analysis are fixed and permeabilized, while live unfixed cells are used for scRNAseq with an additional live/dead cell selection step, and the latter does not include all genome-wide transcripts. We suspect that any differences in populations between the two methods are both a result of differences in sample processing and their own technical limitations, which makes it difficult to directly compare population proportions between the two methods. Because the numbers of CD4 cells were much lower than CD8 cells in both scRNAseq and CytoF, and knocking out CD8 cells abrogated the therapeutic effect of bintrafusp alfa, our data overall support the notion that CD8 T cells as a prominent effector cell type. We have added these points to the revised discussion section (see discussion on lines 367-375).

“As an explanation for their finding that in mice, which are genetically identical, treatment with bintrafusp alfa generates two distinct outcomes, the authors suggest that this may relate to (subtle) differences in T cell clonotype which arise due to random TCR rearrangement in the thymus. Do the authors have evidence, or could they generate such evidence, that there is perhaps a less diverse TCR profile in the non-responding mice compared to the responding mice?”

Response: The reviewer has a valid point. We feel that addressing this question would require a much more thorough study outside the scope of this manuscript. To incorporate the reviewer's point of view, we have put these studies as a future direction in the revised discussion (see discussion on lines 384-385).

“Also do the scRNAseq data on the tumor cells, apart from changes in HLA class I expression, provide insight on whether tumor cell heterogeneity is different between the groups?”

Response: The total number of tumor cells was reduced in responders when compared to non-responders, but that reduction was equal across sub-clusters that made up the “tumor cells” and “dividing tumor cells” clusters. Therefore, it appears that heterogeneity was similar between the groups, although a DNA-based analysis would be a more conclusive way to explore that further. We have added these points to the discussion section (see discussion on lines 379-382).

“Is it possible that epigenetic or metabolism may play a role in these differences: was there a difference in the weight of the mice not responding, which may be related to the fact that they had a different caloric intake?”

Response: There were no differences in mouse weight at the time of both mass cytometry and single-cell sequencing analyses, and any weight differences at later time points was due to tumor burden as mice approached endpoints. We included this point in the results section (lines 223-226) in Supplementary Fig. 8h.

“On p8 an incorrect p-value is mentioned in the text with regards to the inflammatory and resident monocyte infiltration ($p=0.272$), where in figure 4 a p-value of $p=0.0272$ is reported.”

Response: Thank you for pointing this error. This P -value was indeed 0.0272, although P -values have been updated with the pooled experimental results as indicated above.

“Do I understand correctly that for the scRNAseq experimental setup 10,000 cells were captured for each sample and the in total 40,000 cells were sequenced? Or were the samples hashed prior to capturing the single cells and were in total 10,000 cells (mix of all 4 samples) sequenced? This should be described in more detail in the methods. Also, after applying the exclusion criteria as mentioned, how many cells were eventually used (for responder and non-responder samples) for the final analysis?”

Response: We apologize for not clearly describing the method. A target of 10,000 cells were captured per sample, resulting in 40,000 total cells from 4 samples being barcoded and sequenced together in multiplex. After exclusion criteria, 38,491 cells were used for the final analysis. We have clarified this and our final cell numbers used in the methods section (lines 597 and 603-604).

Reviewer #3

“In figure 1a, you measured TGF β in mouse cell lines. Did you measure human cell lines? Also, were those cells (MNK, A223, A1419) with repeated measurement 2 times. So, I would like to repeat more one time.”

Response: We did analyze TGF β production by human cell lines in our ELISA assay, but as this manuscript only uses murine cell lines to be transplanted into immune competent syngeneic recipients in subsequent experiments we opted not to include them, and instead presented the TCGA data to provide clinical context to our human samples. We have increased our replicates to 6 in the repeated ELISA (see Fig. 1a, results on lines 85-88).

“In figure 1b, you checked TGFb1 vs SMAD4 mRNA expression in human cell patient samples. Did you check it in mouse cell lines (A223 cells)?”

Response: A223 cells are Smad4-knockout and do not have detectable Smad4 mRNA expression. We have added this data to Supplementary Fig. 1a and results lines 88-90.

“In figure 1 e-f, tumor growth was inhibited in C57BL/6 mice not nude mice. So, you suggest that systemic TGFβ inhibition improved the effectiveness of T cell mediated immunotherapy. If it is right, did you check in other mice (Balb/c or etc)?”

Response: A223 cells are derived from a C57BL/6 background, so they will not grow in Balb/c mice. We will clarify that those experiments were performed in either syngeneic or immunocompromised recipients (see results on lines 100 and 174-176). Additionally, we have demonstrated the effectiveness of combined TGFβ/PD-L1 inhibition in LY2 cells in a Balb/c background, so that effect is not strain-related.

“In supplement figure 4, you injected LY2 cells in Balb/c mice. If you want to say other mice also affect tumor growth, I think you need to A1419 cells also. Why did you inject LY2 in Balb/c and inject A1419 in C57BL6?”

Response: LY2 cells are derived from a Balb/c background and A1419 cells are derived from a C57BL/6 background. Neither cell line will grow in a non-syngeneic recipient that is capable of mounting an immune-mediated transplant rejection. As above, we have clarified that cell lines were injected into syngeneic recipients in the manuscript, and that the purpose of supplementary Fig. 4 is to demonstrate the effectiveness of bintrafusp alfa in two additional cell lines rather than alternative mouse strains.

Figure 1: Smad4-mutant SCCs in mice and humans exhibit high TGFβ expression, and combined TGFβ inhibition and PD-L1 blockade durably eradicate Smad4-mutant (A223) SCCs.

a pg/mL per cell of secreted TGFβ measured by ELISA in untransformed mouse skin keratinocytes, A223 cells, LY2 cells, and A1419 cells ($n=6$). P -values are shown for multiple comparisons to A223 cells by one-way ANOVA ($F=328.8$, $df=25$).

b *TGFBI* mRNA expression z-scores in human HNSCC patient samples based on their *SMAD4* mutant status of deleted (blue, $n=253$), diploid (green, $n=205$), or gained (orange, $n=30$). P -values are shown for multiple comparisons to diploid cells by one-way ANOVA ($F=8.445$, $df=487$).

c Post-transplant tumor volume of A223 tumors in C57BL/6 mice treated with either 75mg/kg LY2109761 p.o. per day (pink, $n=22$), or carboxymethylcellulose vehicle control (black, $n=22$) for 14 days.

d Tumor volume on day 18 post-transplant of mice in **c**.

e Post-transplant tumor volume of A223 in athymic nude mice treated as in **c** ($n=10$ per treatment group).

f Tumor volume on day 17 post-transplant of mice in **e**.

g Survival post-injection of C57BL/6 mice bearing A223 tumors that were treated as in **c**, with the addition of 200μg anti-PD-L1 or its isotype control 3x/week for two weeks (vehicle/mock: black lines, LY2109761/mock: pink lines, vehicle/anti-PD-L1: blue lines, LY2109761/anti-PD-L1: green lines, $n=9$ per treatment group); surviving mice (combination-treated $n=7$, anti-PD-L1-treated $n=3$) were rechallenged with A223 cells on day 147 (arrow) along with a cohort of naïve mice (dashed line, $n=5$).

h Tumor volume on day 25 post-transplant in mice from **g**. Multiple comparisons were corrected with Tukey's method for **a** and **b**. Unpaired two-tailed t tests were performed for **d**, **f**, and **h**, and survival comparisons were performed using the log-rank Mantel-Cox test. All error bars represent the SEM, except for **b** where the box represents the 25th to 75th percentile, whiskers represent the full range, and the line represents the median value.

Figure 4: Responders to bintrafusp alfa have a more immune-permissive microenvironment and increased T cell activation when compared to non-responders.

a Infiltrating immune cell populations of all mice treated with bintrafusp alfa (green bars, $n=10$), its isotype control (black bars, $n=10$), anti-PD-L1 (blue bars, $n=10$), or TGF β trap (pink bars, $n=10$) after mass cytometric analysis and clustering. **b** Infiltrating immune cell populations of responder (green bars, $n=5$) and non-responder (black bars, $n=5$) subpopulations of bintrafusp alfa-treated mice, with population differences determined as in **a** ($P < 0.0001$, $F=11.17$, $df=10$), with P-values for significant multiple comparisons as shown. **c** PD-1 expression on CD8 T cells from **b** as quantified by mass cytometry ($n=5$). Tumor digests from mice treated as in **a** were treated for 4 hours with brefeldin-A, ionomycin, and PMA and the proportion of **d** CD8⁺IFN α ⁺TNF α ⁺ and **e** CD8⁺Granzyme B⁺ cells are shown ($n=7$ non-responders in black, $n=8$ responders in green). Population differences for **a** ($P < 0.0001$, $F=2.617$, $df=30$) and **b** ($P < 0.0001$, $F=11.17$, $df=10$) were determined by two-way ANOVA and Sidak's multiple comparisons test with significantly altered populations as shown. Unpaired two-tailed t tests were performed for **c-e**, and all error bars represent the SEM.

Figure 7: Anti-CXCR3 abrogates the response to bintrafusp alfa in A223 tumors, resulting in tumors indistinguishable from non-responders to bintrafusp alfa

a Survival post-injection of C57BL/6 mice bearing A223 tumors that were treated with double isotype controls (black line, $n=12$), isotype and bintrafusp alfa (green line, $n=13$), anti-CXCR3 and isotype control (purple line, $n=9$), or anti-CXCR3 and bintrafusp alfa (orange line, $n=10$). **b** Tumor growth in mice treated from **a**. Survival comparisons were performed using the log-rank Mantel-Cox test. Differences between isotype/bintrafusp alfa (green) and anti-CXCR3/bintrafusp alfa (orange) in **b** were identified by two-way ANOVA ($P<0.0001$, $F=3.848$, $df=27$) with multiple comparisons corrected by Tukey's method and indicated by * for $P=0.0328$, ** for $P=0.0028$, and **** for $P<0.0001$. **c** Infiltrating immune cell populations of mice treated with anti-CXCR3 in combination with bintrafusp alfa (orange bars, $n=5$) or its isotype control (purple bars, $n=5$); mice were also treated with bintrafusp alfa alone and separated into non-responders (black bars, $n=4$) and responders (green bars, $n=5$). Total population differences were determined by two-way ANOVA ($P<0.0001$, $F=5.979$, $df=30$) and Sidak's multiple comparisons test with significantly altered populations between bintrafusp alfa responders (green) and anti-CXCR3 in combination with bintrafusp alfa (orange) are shown. Survival comparisons in **a** were performed using the log-rank Mantel-Cox test, and all error bars represent the SEM.

Supplementary Figure 1: LY2 PD-L1 expression requires Smad4, and human HNSCC PD-L1 expression is primarily on stromal cells.

a Smad4 protein (green, 70kDa), and actin protein (red, 42kDa) in HaCaT keratinocytes and A223, LY2, and A1419 tumor cells after western blot analysis. **b** Quantification of phospho-Smad3 (pSmad3) positive cells in A223 tumors 12 hours after treatment with 75mg/kg LY2109761 ($n=5$) or its vehicle control ($n=5$). **c** Representative pSmad3 IHC images of vehicle- or LY2109761-treated tumors quantified in **b**. An unpaired two-tailed t-test was performed for **b**, and all error bars represent the SEM.

Supplementary Figure 5: LY2 and A1419 tumor growth after treatment with bintrafusp alfa.

a Quantification of pSmad3-positive cells in A223 tumors 48 hours after treatment with isotype control (circles), TGF trap (squares), or bintrafusp alfa (triangles) as in **Fig. 3**, and representative IHC staining images for each ($n=4$ per group). **b** Tumor growth of buccal LY2 tumors in balb/c mice treated with isotype control (black circles, $n=6$), TGF β trap control (pink squares, $n=7$), anti-PD-L1 (blue triangles, $n=7$), or bintrafusp alfa (green triangles, $n=7$). **c** Tumor growth of subcutaneous A1419 tumors in C57BL/6 mice treated with isotype control (black circles, $n=7$), TGF β trap control (pink squares, $n=7$), anti-PD-L1 (blue triangles, $n=8$), or bintrafusp alfa (green triangles, $n=8$). Differences between treatments were calculated by 2-way ANOVA using Tukey's correction for multiple comparisons; significant differences between bintrafusp alfa (green) and isotype control (black) are indicated with * for $P=0.0218$, ** for $P=0.0071$, and **** for $P<0.0001$, and all error bars represent the SEM.

Supplementary Figure 8: Individual A223 tumor tracks and size group differentiation for flow cytometric analysis.

Tumor digests from **Fig. 4a** were treated for 4 hours with brefeldin-A, ionomycin, and PMA, and their **a** CD8⁺IFN α ⁺TNF α ⁺ and **b** CD8⁺Granzyme B⁺ cells were quantified by flow cytometry ($n=7$ per group). Individual tumor tracks of mice bearing A223 tumors treated with **c** bintrafusp alfa, **d** isotype control, **e** TGF β trap, and **f** anti-PD-L1 that were harvested for mass cytometric analysis in **Fig. 4** ($n=10$ per group). Tumors were sub-grouped into “large” and “small” size or “responder” and “non-responder” populations as indicated. **g** Differences in average tumor volume for groups identified in **c-f** with significant differences as indicated ($n=5$ per treatment). **h** Mouse weights on mice analyzed in **Fig. 4** and **Supplementary Fig. 8** over the course of treatment ($n=10$ per group). Unpaired two-tailed t tests were performed for **g**. A one-way ANOVA with multiple comparisons corrected for by Tukey’s method was performed for **g** and **h** with significant differences as indicated, and all error bars represent the SEM.

Supplementary Figure 9: Infiltrating immune cell populations do not change based on tumor size or treatment without differentiating between responders and non-responders.

a Infiltrating immune cell populations of large (black, $n=5$) and small (gray, $n=5$) tumor subpopulations of isotype-treated mice from **Fig. 4a** after mass cytometric analysis and clustering. **b** Infiltrating immune cell populations of large (black, $n=5$) and small (gray, $n=5$) tumor subpopulations of TGF β trap-treated mice from **Fig. 4a** after mass cytometric analysis and clustering. **c** Infiltrating immune cell populations of large (black, $n=5$) and small (gray, $n=5$) tumor subpopulations of anti-PD-L1-treated mice from **Fig. 4a** after mass cytometric analysis and clustering. Multiple comparisons were performed by 2-way ANOVA with no significant differences between groups for **a** ($P=0.6885$, $F=0.7363$, $df=10$), **b** ($P=0.2670$, $F=1.258$, $df=10$), and **c** ($P=0.6370$, $F=0.7914$, $df=10$). Post-hoc analyses were performed using Sidak's multiple comparisons test with no significantly altered categories found, and all error bars represent the SEM.

REVIEWERS' COMMENTS:

Reviewer #1 (Remarks to the Author):

The authors have done an admirable job addressing my previous concerns.

Reviewer #2 (Remarks to the Author):

The authors have taken the comments from the reviewers to heart and have provided additional experiments and have enlarged their group sizes in order to provide more robust data sets. Especially the addition of the mode-of-action, showing that CXCR3 blockade can abrogate the therapeutic effect of bintrafusp alfa treatment is a great addition to the manuscript and increases its novelty and impact.

I am satisfied with the changes made to the revised manuscript and the response to my earlier comments. I have no additional comments.